



# Surface current variability in the East Australian Current from long-term HF radar observations

Manh Cuong Tran[1,2], Moninya Roughan[1,2], and Amandine Schaeffer[3,2]

[1]School of Biological Earth and Environmental Sciences, UNSW Sydney, NSW 2052 Australia
[2]Centre for Marine Science and Innovation, UNSW Sydney NSW 2052 Australia
[3]School of Mathematics and Statistics, UNSW Sydney, NSW 2052 Australia

**Correspondence:** M. Roughan (mroughan@unsw.edu.au)

**Abstract.** The East Australian Current (EAC) exhibits significant variability across a wide range of spatial and temporal scales, from mesoscale eddies and meanders to seasonal, interannual, and decadal fluctuations in its intensity, pathway, and influence on the continental shelf circulation. Understanding and monitoring this variability is crucial because the EAC plays an important role in controlling shelf dynamics, regional circulation, coastal weather and global climate patterns. As such, two high-frequency (HF) coastal radar systems have been deployed on the east coast of Australia to measure surface currents upstream and downstream of the East Australian Current (EAC) separation point. The multi-year radar dataset (spanning 4-8 years) is presented here and its use is demonstrated to assess the spatial and temporal variability of the EAC and the adjacent continental shelf circulation, ranging from seasonal to interannual scales. The dataset is gap-filled using a 2dVar approach (after rigorous comparison with the traditional unweighted Least-squares fit (LS) method). Additionally, we explore the representation depth variability of the observations by comparing the data with surface Lagrangian drifter velocities (with and without depth drogues). The multi-year radar-derived surface current dataset, which was validated using short-term drifter and long-term current meter observations, revealed that the local upstream circulation is strongly dominated by the EAC's annual cycle, peaking in the austral summer. The analysis using 8 years of upstream data revealed the period of the EAC intensification at around 3-5 years. The interannual variability of the poleward transport downstream was driven by the intrinsic variability of the jet. This dataset which continues to be collected, complemented by numerical simulations and in-situ measurements, will provide a comprehensive view of the EAC's variability and its impact on the broader regional circulation dynamics which can be used for a range of dynamical investigations.

## 1 Introduction

The southeast Australian coastal zone is home to a diverse array of unique marine habitats and ecosystems and also offers substantial socioeconomic values through various maritime activities and industries. However, understanding and quantifying the complex dynamics that govern these regions poses a major scientific challenge (Roughan et al., 2015). One of the difficulties





is that ocean motions exhibit intricate interactions across a vast range of spatial and temporal scales. This complexity is further amplified in shallow shelf regions, where the currents and circulation patterns exhibit significant variability driven by a wide spectrum of external forcing mechanisms. These give rise to intricate flow patterns and physical processes that are inherently difficult to observe and quantify through conventional means (Simpson and Sharples, 2012).

Owing to this challenge, considerable work has been invested to extend the capacity of the ocean observation network. In recent years, high-frequency (HF) coastal radar has become an important part of coastal ocean observing systems and is recognized as an efficient tool for studying and monitoring coastal regions (Roarty et al., 2019). HF radar is a remote sensing technique that measures surface ocean currents and waves from the shore. The HF radar interprets surface currents by analyzing the backscattering of radar-emitted signals, known as 'backscatter'. These backscatters are induced by the surface ocean ripples from long wind-generated waves (with wavelengths ranging from 3 to 30 meters) in the ocean. Based on analyzing the spatially and temporally varying radar signal, information on the sea surface wind, waves, and currents can be obtained (Paduan and Washburn, 2013). The advantage of the HF radar comes from its ability to continuously monitor surface currents and waves at high frequency across a wide area up to hundreds of kilometers offshore. This advantage of HF radar plays a crucial role in improving the observation capacity. It effectively bridges the gap between continuous and local measurements obtained through in-situ methods (such as mooring observations) and the broader but less frequent satellite data. By combining radar measurements with other techniques, the HF radar data provide a comprehensive description of surface currents, from hours to interannual variations.

By way of example, several studies have occurred in recent years using HF radar, such as the analysis of long-term variation of Soya Warm currents (Ebuchi et al., 2009), seasonal shifts of the western United States coastal shelf circulation (García-Reyes and Largier, 2012), the variability of the east Australian current (Archer et al., 2017a), Florida current (Archer et al., 2017b), the variability of Gulf streams (Muglia et al., 2022), and comparison studies between systems (Archer et al., 2018). Other applications include focusing on dynamic features such as small eddies (Mantovanelli et al., 2017; Schaeffer et al., 2017), marine renewable energies, and wind/wave interaction (Ardhuin et al., 2009; Dzwonkowski et al., 2009; Thiébaut and Sentchev, 2016; Schaeffer et al., 2020) inter alia.

Off the east coast of Australia, the East Australian Current (EAC), a highly dynamic western boundary current of the South Pacific Subtropical Gyre, plays an important role in the marine ecosystem and climate of the region (Fig. 1a). It redistributes heat, marine organisms, nutrients, and debris while moderating weather patterns and climate dynamics by transporting warm subtropical waters poleward toward the temperate mid-latitudes. On a local scale, the EAC significantly influences shelf dynamics in multiple ways (Schaeffer et al., 2014, 2017; Malan et al., 2023). Originating between 10-20°S, the EAC strengthens, meanders consistently, and flows southward along the coast, carrying an average transport of about 22 Sv. It eventually separates at around 30-32°S, transitioning into an eastward flow known as the Tasman front and a field of southward propagating eddies extends poleward (Oke et al., 2019). According to the literature (Kerry and Roughan, 2020; Cetina-Heredia et al., 2014), the EAC intensifies and separates from the coast between 31-34°S and between 32-33.5°S 38% of the time specifically. The EAC remains attached to the coast from its origin around 18°S until about 32°S, where part of the jet turns eastward as the Tasman Front towards the subtropical gyre of the South Pacific and another continues southward as the EAC southern exten-





sion, linking the three major Southern Hemisphere gyres (Oke et al., 2019). The southern extension pathway of the EAC can be recognized from the multi-year mean circulation pattern in Fig. 1a, which shows the poleward velocity at NEWC is half as strong as the velocity at COF. The EAC, like other western boundary currents, plays a crucial role in the oceanic circulation system.

In this regard, the HF radar has been deployed as part of the effort to monitor the East Australian Current (EAC). HF radar was first deployed off southeastern Australia off Coffs Harbour in March 2012 by Australia's integrated marine observation system (IMOS, Table 1). Since then, HF radar observations have been used to advance our understanding of the regional dynamics and variability of the EAC (e.g., Archer et al. (2017a); Mantovanelli et al. (2017); Malan et al. (2023); Schaeffer et al. (2017), etc.). Flowing near the narrow continental shelf, the EAC interacts with coastal topography, enhancing uplift and upwelling processes that replenish nutrients and maintain high biological productivity (Roughan and Middleton, 2004). The proximity of the EAC to the shelf creates intricate structures such as frontal eddies and density fronts (Mantovanelli et al., 2017; Schaeffer et al., 2017; Bourg et al., 2024). These short-lived dynamic features are often associated with large horizontal and vertical velocities and thus strongly influence mass transport (D'Asaro et al., 2018). Along with the southward movement, consistent meandering of the EAC on- and off-shelf causes a large volume of cross-shelf exchange, (up to 3.5 Sv) (Malan et al., 2022), with higher variability downstream of the separation point related to eddy-shedding and interactions (Malan et al., 2022). Furthermore, recent studies have shown that the EAC's increased poleward penetration leads to an increase in eddy activity and more warm water being transported toward the south, thereby contributing to the Tasman Sea's warming trend and affecting broader climate patterns (Cetina-Heredia et al., 2014; Li et al., 2022). Indeed, these dynamic features induced by the EAC directly impact the circulation, biological production, marine ecosystems, and fisheries of the eastern Australian continental shelf.

HF radar measures the radial component of surface currents and to resolve a total current vector, two or more stations are normally required. The unweighted Least-squares fit approach (LS) is the most commonly used method to combine radial observations into a current vector (Wyatt et al., 2018). This technique aims to minimize the error between radial velocities by applying a uniform weight coefficient to all observations. It has been implemented by the IMOS radar team to create the initial version of the radar surface current dataset (Cosoli and Grcic, 2019; Wyatt et al., 2018). Despite its simplicity, surface current velocities processed by the LS method are prone to inaccuracies due to reduced radial coverage, resulting in a decrease in the total surface currents (Fig. 1b). Additionally, the radar observations can be affected by several factors ranging from environmental interference (sea state conditions, ionospheric disturbance, etc.) to technical failures (Liu et al., 2014), leading to the data loss and reduction of data accuracy (2).

The variational approach (2dVar) proposed by Yaremchuk and Sentchev (2009), offers a method to obtain accurate current velocity maps over extended periods. This non-local, kinematic-constrained interpolation technique overcomes the limitations of the least-squares (LS) method by utilizing all observational points to produce continuous, gap-free datasets. This feature of the interpolating technique can help to overcome some limitations related to a lack of data, which frequently occurs in radar measurements. Unlike the LS method, which struggles with data gaps or discontinuities, the 2dVar approach provides a more comprehensive solution for ocean current mapping (Yaremchuk and Sentchev, 2009). The 2dVar method has been successfully





utilized and demonstrated an outstanding performance for reconstructing the radar-derived surface velocity in other data sets, e.g. Bodega Bay (Yaremchuk and Sentchev, 2009, 2011), the Iroise sea (Thiébaut and Sentchev, 2016) and the Gulf of Tonkin
(Tran et al., 2021). Here we compare these two methods and their ability to handle incomplete or gappy data and to generate continuous current velocity maps to provide a comprehensive and gap-free dataset.

High-frequency (HF) radar systems, which detect signals scattered by surface waves, can only measure currents in the top layer of the ocean. The effective depth of these measurements can be identified by the properties of surface gravity waves by using a formula $d = \lambda/(8\pi)$ (Stewart and Joy, 1974). Factors such as surface stress, wave action, and stratification can
alter the current profile in the upper water column, potentially creating discrepancies between different measurement methods and affecting the accuracy of velocity estimates derived from radar data. The measurements of vertical shear in the uppermost meter of the wind-influenced ocean surface are challenging to obtain, largely due to technical constraints of current instruments (Lodise et al., 2019). Various studies have attempted to validate HF radar measurements against other instruments, including drifters and ADCPs (e.g., Sentchev et al. (2017); Wyatt et al. (2018); Molcard et al. (2009); Rypina et al. (2014); Dumas et al.
(2020); Capodici et al. (2019)). However, a definitive conclusion regarding radar uncertainties remains elusive. This lack of consensus is primarily due to the challenges in comparing instruments that measure at different depths. To explore these issues, we compare the HF radar data with current velocity estimates from surface Lagrangian drifters that measure at different depths through the water column. By analyzing data from a selected group of these drifters, we aim to assess how vertical shear in the near-surface layer influences the uncertainty in HF radar measurements.

In this study, we describe the HF radar systems in detail (Section 2) and provide comprehensive metadata for the ongoing use of the data. In Section 3 we describe the data availability, and the data quality control (QC). We do a rigorous comparison of 2 different methods to reconstruct the surface velocities (LS and 2dVar) and provide a gap-filled dataset. We validate the data set using velocity estimates from drifters representing the water at 3 different depths (drogue and undrogued drifters) and current meter moorings, showing the depth of the HF radar observations. Using the gap-filling method we construct and validate a
novel multi-year radar dataset that is useful for studying the dynamics of the East Australian Current. In Section 5 we present new insights into the variability of the EAC System using the dataset and discuss the limitations of the HF radar observations.

## 2   The HF radar network along southeast Australia

Along southeastern Australia, the HF radar systems are operated as part of the IMOS radar network, to enhance observations and understanding of the ocean around Australia. HF radar has been operational along the southeastern coast since 2012, acting
as a supplementary observation platform to the other IMOS infrastructure such as moorings (Roughan et al., 2015). In the EAC the HF radar network currently consists of two radar sites one located around Coffs Harbour ( 30°S) (COF) and the other around Newcastle ( 32°S) (NEWC). Both radar systems overlook the surface waters of the continental shelf off the eastern Australian coast and the EAC, allowing for monitoring and assessing the intricate details of the EAC's behavior and its impact on the shelf environment.



The first system at Coffs Harbour (COF) in the north has two WEllen RAdar-manufactured (WERA), standard-range radars, operating at the frequency of 13.5 MHz and the bandwidth of 100 kHz. Two land-based WERA radars have been deployed at Red Rocks (RRK) and North Nambucca (NNB) to observe the upstream of the EAC separation region (Fig. 1a). All the metadata associated with the HF radar systems are shown in Table 1. This includes information on the bandwidth, azimuthal resolution, observation range, radial resolution, temporal resolution, data coverage and relevant published literature. Also

shown are changes to the configurations over time. The radar data are processed using standard WERA software onto a rectangular grid with a horizontal resolution of 1.5 km. The radial data is provided every 10 minutes and has an operational range of up to 150 km. The hourly radial data from two COF radar sites were merged by a 5-point moving average (Wyatt et al., 2018). The radar wavelength, computed as $c/f$ (where c is speed of light), is 22.2 m. The average depth of current measurement can be identified based on the relationship with the transmitted radio wavelength, $\lambda/8\pi$ (Stewart and Joy, 1974), which is roughly

0.9 m.

In late 2017, the second HF radar system was deployed that observes the Hawkesbury Shelf Region lying immediately downstream of the typical EAC separation zone off Newcastle (NEWC). Two Seasonde radars have been deployed at Seal Rocks (SEAL) and Red Head (RHED) (Fig. 1a), providing hourly data (Table 1). The system comprises two long-range radars,

manufactured by CODAR Ocean Sensor (CODAR), operating at 5.3 MHz with a range of up to 200 km and a horizontal resolution of 5.8 km. The radar wavelength is 56.6 m, giving an average depth of 2.3 meters. Shortly after its deployment, the NEWC radar was shut down for about a year due to radio interference. The system resumed operations in 2018, along with a significant reduction in transmitted power (less than 1 W) and bandwidth (reduced from 26kHz to 14kHz) (Cosoli, 2020). The development and implementation of a "listen-before-talk" mode (Cosoli, 2020) and an adjusted bandwidth mitigated the

interference with local operations, but this also resulted in reduced spatial resolution and observation range (from 200 km to 100 km). All the metadata and the changes to the system are shown in Table 1).

The CODAR and WERA radar systems have different ways of determining the signal direction. As the transmitted radio wave is reflected to the radar from every direction, the WERA-manufactured radar relies on the beam-forming method to determine the direction which the signal comes from. The beam-forming method requires the receiver antennas to be arranged

in two separate arrays with a rectangular shape (transmitter array) and a linear or curvilinear array of receive antennas composed of 16 elements with the maximum azimuthal spreading of 120° for the system in COF region (Wyatt et al., 2018). The CODAR system, on the other hand, uses a three-element antenna, of which are perpendicular to each other, in one receiver box and determines the direction of arrival signals by the method of directional finding (Cosoli and Grcic, 2019). At all four sites, the radar system has been routinely maintained and calibrated as part of the IMOS network following best practice.





**Table 1.** Summary of the metadata for the two radar sites and their associated configurations and capability including changes to the system

| Site | Coffs Harbour (COF) | | Newcastle (NEWC) | |
|---|---|---|---|---|
| | Red Rock (RRK) | North Nambucca (NNB) | Seal Rocks (SEAL) | Red Head (RHED) |
| Location | 29.98°S, 153.23°E | 30.62°S, 153.01°E | 32.44°S, 152.54°E | 33.01°S, 151.73°E |
| Start date | 03/2012 | 03/2012 | 11/2017 | 11/2017 |
| Center Frequency | 13.92 MHz | 13.92 MHz | 5.3 MHz | 5.3 MHz |
| Bandwidth | 100 kHz | 100 kHz | 26 kHz (before 09/03/2018) 14 kHz (09/03/2018 - present) | 26 kHz (before 09/03/2018) 14 kHz (09/03/2018 - 18/07/2019) 11 kHz (18/07/2019 - present) |
| Azimuthal Resolution | 10.36˚ | 10.36˚ | 5˚ (before 13/06/2019) 2˚ (13/06/2019 - present) | 5˚ (before 31/07/2019) 2˚ (31/07/2019 - present) |
| Observational Range | 150 km | 150 km | 150 km | 150 km |
| Radial Resolution | 1.5 km | 1.5 km | 5.8 km | 5.8 km |
| Temporal Resolution | Raw data: 10-minute QC data: 1-hour | Raw data: 10-minute QC data: 1-hour | 1-hour | 1-hour |
| Temporal coverage | 03/2012 - present | 03/2012 - 06/2021 | 11/2017 - present | 11/2017 - present |
| Estimated depth of measurements ($\lambda/8\pi$) | 0.9 m | 0.9 m | 2.3 m | 2.3 m |
| References | Archer et al. (2017b, 2018); Schaeffer et al. (2017); Mantovanelli et al. (2017); Wyatt et al. (2018); etc. | | Cosoli (2020); Malan et al. (2023); Bourg et al. (2024) | |

## 3 Data and methods

### 3.1 Data quality control and availability

*HF radar radial velocities*

The radial data availability for each site from March 2012 to January 2024 is shown in Fig. 2. Radial data from all sites are freely available and assessed through the IMOS data server (https://thredds.aodn.org.au/thredds/catalog/IMOS/ACORN/catalog.html). Inaccuracies in calculating the radial currents can arise from various factors, including radio-wave interference from moving ships, misidentifying of the Bragg peaks or the directional of arrival due to inadequate calibration of the antenna pattern and additional uncertainties introduced during the vector mapping process (Wyatt et al., 2018). An IMOS standard

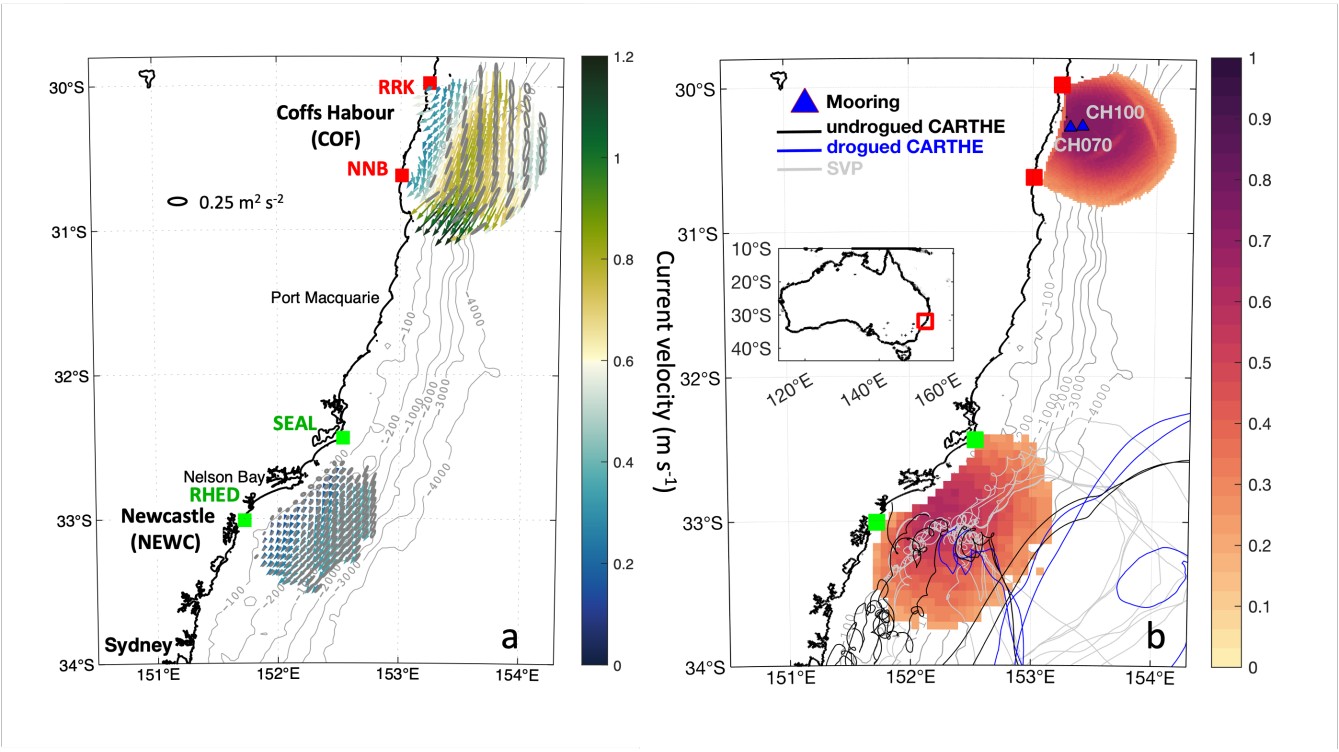

**Figure 1.** Maps showing the location of the radar sites along the east coast of Australia. (a) Time mean surface current vectors observed from Coffs Harbour (COF, RRK + NNB sites) (2012-2021) and Newcastle, (NEWC, SEAL + RHED sites) (2018-2023) radar sites. The ellipses denote the current velocity variance (plotted every 2 grid points). The color map depicts the mean velocity magnitude. (b) Map of the mean spatial coverage (as a ratio from 0 to 1) for the two sites; COF (2012 - 2021) and NEWC (2018 - present), where data were averaged for a period of 8 years for COF and 4 years for NEWC. The metadata for each radar site can be found in Table 1. Also shown are the locations of the two sub-surface current meter moorings at Coffs Harbour (Ch070 and CH100) and the trajectories of the surface drifters used for validation of the NEWC system as described in Section 3.1.

quality control procedure was applied to remove the data outliers from the original radial data (FV00). The data with quality control (QC) is flagged as Level 1 (FV01) and is published to the AODN server at a delay of a few months. We refer the readers
to Cosoli (2020) for more information on the QC protocol. At the time of this study, the QC'd data is available for the COF region from 2012 to 2021, however, the QC'd data for the NEWC region is available for the total current vectors, while for the radials, it only lasted for about two years, from May 2018 to October 2019. Therefore, to acquire better data, further QC was applied to the FV00 radial data at the NEWC region. Here, we followed the method of Bourg et al. (2024) for removing outliers in the radial dataset which is based on Cosoli and Grcic (2019). The outliers in the radial velocity were identified based on the
absolute velocity exceeding 3 standard deviations across the entire time series as well as pixels with less than 30% temporal coverage during a month. Then, the spatial and temporal gradients of velocity were examined to eliminate the data points with a probability of occurrence below 3%. Finally, the gradient of absolute current speed for each grid cell over the whole dataset

Earth System
Science
Data

Open Access / Discussions

exceeding 3 standard deviations within a moving window of 30 points was considered a spike and removed. This procedure
is iteratively repeated at every data grid cell to ensure quality across the entire dataset, leading to a cleaner and more robust

dataset suitable for further analysis and interpretation.

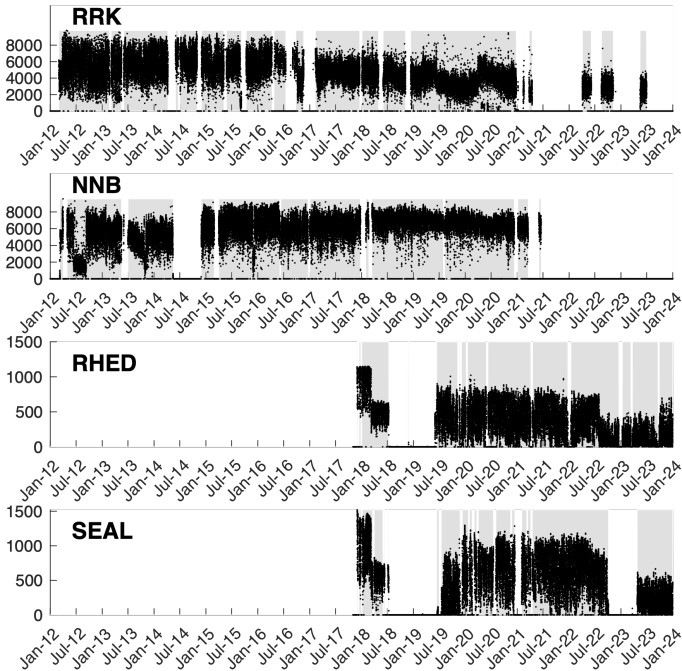

**Figure 2.** Temporal coverage of the data available at each radar site to date. Black points present the coverage of hourly radar data for each
site. Gray shading represents the radar in operation whereas the blank spaces show when the radar stopped working for more than 1 day.

*Current meter mooring data*

Two bottom-mounted acoustic Doppler current profilers (ADCPs) are located at the 70 m (CH070) and 100 m isobaths
(CH100) across the shelf around 30°S (under COF radar coverage, Fig. 1a), in the middle of the COF domain provid-
ing current observations at high temporal resolution through the water column. The velocities were monitored with verti-

cal bin sizes of 4 m and sampling at the rate of 5 min. The data were quality controlled using the IMOS Toolbox (http:
//code.google.com/p/imos-toolbox) before being averaged hourly as described in Wood et al. (2016). The topmost bin of the
velocities within the range from 9 to 11 m depth was used in this study. During the study period, mooring data were available
for more than 90% of the time (see Fig. A1). For more information on the mooring data, the readers are referred to Schaeffer
et al. (2014); Roughan et al. (2015).

*Surface and subsurface drifters*

Here we use data from various drifter types, which included the Consortium for Advanced Research on Transport of Hydro-
carbon in the Environment (CARTHE) drifters, both with and without drogues, having drafts of 60 cm and 5 cm respectively,
and the Surface Velocity Program (SVP) drifters at 15 m depth. These drifters were released near the Newcastle (NEWC) HF





radar in two separate campaigns conducted in November 2020 and October 2023. The drifters will be described more in detail
in Section 3.1. During the study period, 20 drifters deployed within the NEWC radar domain were collected, consisting of 7
surface CARTHE drifters and 13 subsurface Surface Velocity Program (SVP) drifters (Lumpkin et al., 2017). The CARTHEs
are donut-shaped, cost-effective, biodegradable drifters that aim to quantify the current transport and material dispersal (such as
oil spills, pollutants, marine debris, etc.) at the ocean surface. The flat design of the drifter is specifically tailored for tracking
surface transport to a depth of 60 cm with the aid of a drogue. Their position is transmitted every five minutes through the
Iridium satellite network. Along with the drogued CARTHEs, two CARTHE drifters without the drogues (drifting at about
5 centimeters) were also deployed to assess the effect of drifter slip velocity introduced by the wind and waves (Novelli et al.,
2017).

The SVP drifter measures surface currents and other oceanographic parameters in the global oceans (Lumpkin et al., 2017).
It is an important tool for studying ocean circulation patterns, understanding the role of currents in global climate, tracking
pollutants, and monitoring marine ecosystems. During the years 2020 and 2023, 13 SVP drifters were deployed over the
Hawkesbury shelf (within the coverage of NEWC radar) to track the near-surface current and transport. The SVP drifter was
equipped with a holey-sock drogue centered at a depth of 15 m which helped to minimize the influence of wind and waves
on the drifter's motion, allowing it to follow the ocean currents more accurately. The data were transmitted every hour. The
availability of the drifter data is shown in Fig. A1.

In order to compare the drifter velocities with the radar-derived velocities, the drift velocities are computed by the finite
difference method along their trajectories on a regular time interval (1 hour) to match the radar time resolution. The HF radar
velocity in the closest cell to the drifter position is interpolated onto the drifter trajectories for comparison. The QC of the
drifter to eliminate inaccurate GPS fixes was done as follows. Every drifter distance smaller than the GPS error (approximately
10 m) was removed from the dataset. Any drift speed exceeding 3 ms$^{-1}$, calculated using finite differencing from the drifter
data, was considered a spike and was filtered out. In addition, we applied a 6-hour Gaussian filter window across the drifter
speed to identify the trends following the initial spike removal. The discrepancies between the filtered and the raw data were
assessed, allowing for the detection of abrupt changes by setting acceptable gradient thresholds. The data points were discarded
if the drift speed surpassed the threshold higher than one standard deviation of the original drift speed.

*Wind data*

In this study, we use wind data from the high-resolution regional reanalysis dataset BARRA2 (Su et al., 2022), provided by
the Bureau of Meteorology (BOM) in Australia. This choice is due to the limited availability of wind stations in our study
region, which are mostly confined to coastal areas. The BARRA2 reanalysis data represents the second reanalysis version from
the Bureau, featuring an enhanced spatial resolution of 12 km and covering Australia and the surrounding regions. The data is
available from 1979 to present day and is provided on an hourly basis (Su et al., 2022).

## 3.2 Reconstruction of the surface current vectors

In principle, HF radar measures the surface currents using wavelengths that interact with surface gravity waves whose prop-
agation is affected by currents at depths of one to several meters ("Bragg scattering"). One radar can only measure the radial



current velocity, which means the currents coming inward or outward of the radar along the radial beams. Therefore, having a full picture of the ocean currents requires two or more radars with a common overlapping zone for completing a surface current
map or the total currents.

While the IMOS radar team provides hourly current velocities interpolated using the LS approach, here we compare the IMOS data with the reconstructed data processed from the variational interpolation (2dVar) approach (Yaremchuk and Sentchev, 2009) which is thought to produce a "best" velocity field $v$. The method is based on the maximum likelihood estimation of its Gaussian probability density function within a predefined oceanic domain $\Omega$. The goal is to obtain the gridded
velocity field $v(x,y)$ at every time step $t$. With 2dVar, a cost function, $J$, is written in a quadratic form and consists of two arguments. The first term in the Eq. 1 involves in the minimization of the error between the unknown velocity $v$ and the radar observations $v_k$. The error is scaled by the variance of the radar measurements at each point, $\sigma^2(v_k)$. The unknown velocities, $v$, are projected along each radial beam $r_k$ with the bearing angle $\theta_r$ to derive a set of discrete data points at location $x_k, k = 1, 2, ...K$. In the second term, the algorithm enforces the pattern of the velocity field via regularizing the spatial deriva-
tives of $v$, which are the divergence, $curlv = \partial_x v - \partial_y u$, and convergence of the velocity field, $divv = \partial_x u + \partial_y v$ within the domain bounded $\Omega$. The smoothness parameters, $W^d$ and $W^c$, in the Eq. 1, are introduced at every grid point to facilitate the extraction of the large-scale circulation pattern while limiting the generation of spurious small-scale variations in the reconstructed velocity field. Similarly, the last parameter, $W^u$, is introduced as an additional smoothness of the velocity field $v$, which was argued by Yaremchuk and Sentchev (2009) to enforce the coherence of the reconstructed velocity field. This
approach maintains the extraction of important physical features in the flow field while reducing the number of artifacts from the interpolation. Surface velocities at the two sites, constructed using the two methods are shown in Fig. 3. Compared to the surface current field generated by the LS approach (Fig. 3a,c), the spurious vectors lying at critical Geometric Dilution of Precision (GDOP) regions are well reduced in the current field processed by the 2dVar (Fig. 3b,d).

$$J = \frac{1}{2K} \sum_{k=1}^{K} \sigma^{-2}(v_k)[(\hat{P}_k v).r_k - v_k]^2 + \frac{1}{2A} \int_{\Omega} [W^d(\Delta divv)^2 + W^c(\Delta curlv)^2 + W^u(\Delta v)^2]d\Omega \qquad (1)$$

Where the K is the number of radar observations, A is the area of the interpolation domain $\Omega$ and $\hat{P}_k$ is the projection of the reconstructed velocity $v$ onto the radial beam at position $x_k$.

### 3.3  Methods of analysis

*Assessing the performance of the 2dVar approach*

By using the 2dVar approach, we can fill the gaps in the reconstructed field, making it crucial to evaluate the performance of
the method. For quantifying the accuracy of the 2dVar method, we employed cross-validation for comparison (e.g., Alvera-Azcárate et al. (2005)). Some cross-validation points were chosen arbitrarily from 1160 radial current snapshots for NEWC radar and 1189 snapshots for COF radar with the best coverage. From each site, roughly 2% of the total radial points were marked for validation and these points were not used in the subsequent analysis. The remaining dataset was used to compute

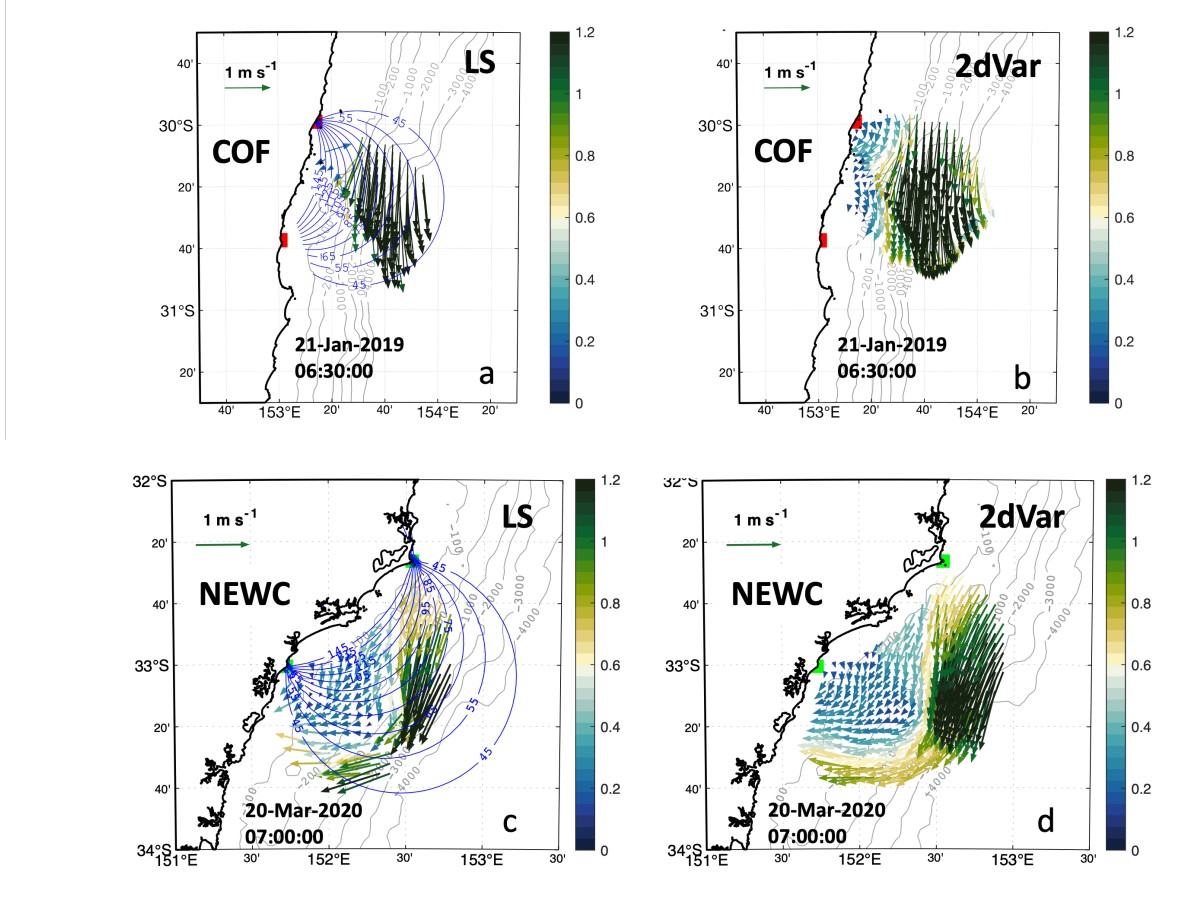

**Figure 3.** The snapshots of surface current fields reconstructed by the traditional unweighted Least-squares fit (LS) (a, c) and by the 2dVar (b, d) methods for two radar sites. The vectors in COF are plotted every 3 grid points for visualization. The white band from the velocity color bar represents the speed threshold of 0.6 m s$^{-1}$. The blue contour shows the radial beams intersecting angle (Geometrical dilution of precision, GDOP). The current vectors are strictly between 45° and 145° of the GDOP.

the total current vectors by the 2dVar method. These analyzed current vectors were then interpolated onto the locations corre-

sponding to the independent validation dataset, facilitating the evaluation of the accuracy of the analysis. Synthetic gaps were introduced into the original velocities covering 10% ($\xi = 0.1$) to 50% ($\xi = 0.5$) of the total grid. The total surface velocities after reprocessing by the 2dVar approach were projected onto the radial beam, $u_r = u_m \cos\theta_r + v_m \sin\theta_r$. These points were then compared with the cross-validation points set aside. Details of the comparison between the reconstructed velocities with gaps and the original data are shown in Table 2.

In order to compare the radar-derived velocity with the in-situ measurements from mooring and drifter data, we adopted some common statistical metrics as used in previous studies e.g. Liu et al. (2014). When comparing two scalar time series, the correlation coefficient ($r$) is a widely used statistical measure to quantify their agreement. This quantity is a statistical measure


that quantifies the strength and direction of the linear relationship between two variables. Additionally, we used the mean bias error to qualify the misfit between the interpolated velocity $X_m$ and the raw data $X_m^*$.

$$Bias = \langle |X_m - X_m^*| \rangle \tag{2}$$

The brackets $\langle ... \rangle$ represent the average over time. In practice, the Root Mean Squared error (*rmse*) is also commonly used to measure the differences between predicted values and observed values as the *rmse* takes into account the mean error of the distance between two time-series.

$$RMSE = \sqrt{\frac{1}{N} \sum_{t=1}^{N} (X_m(t) - X_m(t))^2} \tag{3}$$

The last metric we used for the comparison is the complex correlation between two-time vectors (Kundu, 1976). The complex correlation coefficient (CC) provides a way to measure the full correlation between the two vectors, including both the amplitude correlation $\alpha$ and the phase angle displacement $\theta$.

$$\alpha = \frac{\langle u_o u_m + v_o v_m \rangle + \langle u_o v_m + v_o u_m \rangle}{\sqrt{\langle u_o^2 + v_o^2 \rangle} \sqrt{\langle u_m^2 + v_m^2 \rangle}} \tag{4}$$

$$\theta = \arctan \frac{\langle u_o v_m - v_o u_m \rangle}{\langle u_o u_m - v_o v_m \rangle} \tag{5}$$

Where $(u_o, v_o)$ and $(u_m, v_m)$ represent the zonal and meridional components of the current from observations and HF radar, respectively.

*EAC detection from HF radar*

We used the COF radar data to detect the variability of the EAC following the algorithm of Archer et al. (2017b, a) and
summarized as follows. To set up a coordinate system that moves along with the meandering jet stream instead of being fixed to geographic locations, we first need to identify the central core of the jet at each latitude. We do this by finding the location of the maximum southward velocity for each row of data going from south to north of the COF radar domain. Next, we calculate the general downstream direction that the jet core is traveling at each of those central core points. Then for each core point, we find all the data points that lie along a line cutting perpendicularly across the jet core at that location. For each of
these perpendicular points, we calculate its cross-stream distance away from the core. We also rotate its original east-west and north-south velocity components to make one component perpendicular to the core (the cross-stream component) and the other component along the core's downstream direction. Finally, we binned and re-gridded all the data points using the cross-stream distance from the core as the cross-stream coordinate, and the latitude of the core point as the along-stream coordinate. This allows us to view the jet structure in a coordinate system that moves along with the meandering jet, rather than being tied to
fixed geographic locations.





*Time series analysis*

Spectral analysis is employed to evaluate the intensity of diverse periodic signals within the data, ranging from tidal to inter-annual timescales. Utide tidal harmonic analysis was used to analyze the tidal currents (https://www.po.gso.uri.edu/~cod iga/utide/utide.htm). The Utide toolbox provides a robust algorithm for tidal harmonic analysis to extract tidal constituents from observed data, allowing to handle data with gaps and outliers (Codiga, 2011). To evaluate the variability in low-frequency bands, we applied a fourth-order Butterworth low-pass filter to eliminate short-term fluctuations. In this study, the filtering was conducted with a 25-hour cutoff frequency to remove tidal currents, a 30-day cutoff to filter out seasonal changes, and a 1-year cutoff to identify inter-annual variations in the radar-derived surface circulation. This allows us to isolate and study the different physical processes driving the ocean currents.

## 4 Results

### 4.1 Error analysis of the reconstructed velocities

#### 4.1.1 Performance of the gap-filling by 2dVar

The RMSE discrepancy of the current fields reconstructed by the 2dVar method relative to the validation points is shown in Fig. 4. The accuracy of the resultant velocity field is related to the availability of the data, the accuracy of measurements, and the intersection of the radial beams (GDOP). It is shown that resultant velocity errors are within 5 - 7 cm s$^{-1}$ for the majority of the domain (Fig. 4). Note that higher velocity errors are found off the region of the SEAL radar site toward the offshore region. This higher discrepancy offshore is probably linked to poor coverage of the radar measurement at the outer bound region and the imbalance of radar observations. The skill score for each of the synthetic gap scenarios is shown in Table 2. For more than 20% ($\xi > 0.2$) of cutoff data in the domain, the reconstructed velocity map is degraded quickly with the mean error exceeding more than 50% of the "true" velocity. For more than 30% ($\xi > 0.3$) of the data cutoff, the 2dVar struggles to reproduce the complex current fields in which the finer-scale motions are lost and the current field becomes smoother with the increased number of gaps (not shown). Indeed, this result is quite similar to the experiment in Bodega Bay (Yaremchuk and Sentchev, 2009) where the authors suggested that 80-90% of observational points are required to acquire the most accurate velocity field.

#### 4.1.2 Comparison with in-situ and drifter velocity data

*COF radar and mooring comparison*

The COF radar data was compared with data from two current meter mooring stations (CH070 and CH100) while the data from NEWC was assessed with the drifter data (noting there were no moorings at NEWC, and no drifters at COF). The hourly radar-derived total velocities at COF from July 2012 to October 2020 were bi-linearly interpolated to the two mooring stations. Here, we compare two methods for reconstructing the total velocities (LS and 2dVar). The comparison between the radar-derived and topmost bin velocities of the moorings (approximately 9 to 11 m depth) was made using the common metrics including complex correlation, phase difference, bias, and root-mean-squared errors (Table 3) over about 8 years of data. We found that the

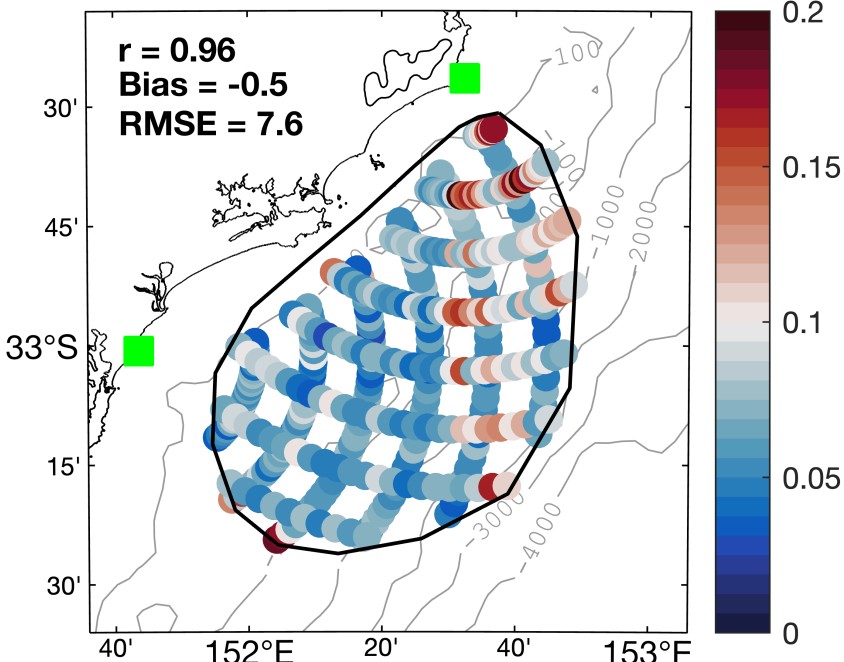

**Figure 4.** The RMSE discrepancy between the NEWC radial measurements and the 2dVar projection of the gap-filled velocity at the cross-validation points (units in m s$^{-1}$). The black contour line delineates the boundary of the analysis domain, selected to encompass 80% of the radar coverage.

**Table 2.** Comparison between the reconstructed data and the withheld data made at the cross-validation points. The radar velocities after reprocessing by the 2dVar approach were re-projected onto the radial beam. The $\xi$ represents the level of gaps within the initial data from 10% ($\xi = 0.1$) to 50% ($\xi = 0.5$).

|  | Level of gaps | $\xi = 0$ | $\xi = 0.1$ | $\xi = 0.2$ | $\xi = 0.3$ | $\xi = 0.5$ |
|---|---|---|---|---|---|---|
| **COF** | $r$ | 0.99 | 0.97 | 0.95 | 0.89 | 0.74 |
|  | Bias ( cm s$^{-1}$) | 0.03 | 1.7 | 2.7 | 4.7 | 9.0 |
|  | RMSE ( cm s$^{-1}$) | 3.4 | 5.4 | 8.0 | 12.6 | 21.9 |
| **NEWC** | $r$ | 0.96 | 0.93 | 0.89 | 0.86 | 0.78 |
|  | Bias ( cm s$^{-1}$) | -0.5 | -0.3 | -0.3 | 0.7 | 1.9 |
|  | RMSE ( cm s$^{-1}$) | 7.6 | 9.0 | 10.9 | 12.2 | 15.9 |





HF radar velocities in the upstream region, which correspond to a depth of 0.9 meters, showed a closer correlation with the in situ data collected at the CH100 site ($\sim 0.89$) compared to the CH070 site ($\sim 0.73$) and a larger cross-shore difference ($\sim 0.20$ m s$^{-1}$) (Table 3). Furthermore, the comparison of the v-component (north-south, alongshore direction) from the mooring and

the radar velocities from both methods (LS and 2dVar) also indicate a higher correlation than for the u-component (east-west, cross-shelf direction) (Table 3). Applying a 25-hour low-pass filter to both data as was done in Wyatt et al. (2018) to remove the high-frequency variation, we found a slightly higher complex correlation $\alpha$, ranging from 0.78 to 0.93 and the RMSE reduced to about 0.16 m s$^{-1}$. The $\theta$ values derived from both LS and 2dVar methods were indeed similar, showing a more clockwise rotation of the subsurface mooring data compared to the surface radar-derived current vectors. A small discrepancy

in angle of around $1°$ was found between the CH100 mooring location and the radar measurements suggesting the subsurface current vectors are nearly aligned with the surface radar vectors. The presence of the EAC likely unified the dynamics from the surface to deeper levels, which was shown to be present approximately 79% of the time (Archer et al., 2017a). Despite a high correlation between the two data sources, a large discrepancy was particularly evident in the CH070 mooring data. The likely cause of this discrepancy can be attributed to the mooring's proximity to the coastline and the baseline between the two radar

sites. The radar-derived velocity measurements at this location may have been compromised by interference from the antenna sidelobes or small-scale disturbances close to the coast as suggested by Wyatt et al. (2018), leading to potential contamination or distortion of the u-component.

*NEWC radar and drifter comparison*

Drifters were within the domain of the NEWC radar coverage for a total of 44 days (Fig. A1). In general, the examination of the NEWC radar-derived total velocities with the drifter velocities for two deployments (from 10 November 2020 to 13 November 2020 and from 10 October 2023 to 15 October 2023) showed the radar-derived velocities compared well with drifter velocities. The comparison between the radar and drifter velocities was made within the timeframe that the drifters were present inside the radar domain (Table 4). For an example of the deployment during November 2020, there were around 3 days of data and

even less of around $\sim 1$ days for the group of drifters released close to the coastline (Fig. 5b, c). In October 2023 the drifters were released in the middle of the radar domain (not shown), whereas in November 2020, drifters (composed of the undrogued CARTHE and the SVPs), were deployed in two separate locations. One group was close to the shore while the others were deployed about 20 km offshore. The time evolution of the drifter and the radar-derived current vectors indicated a reversal of CARTHE drifter vectors to radar-derived current vectors (Fig. 5b, d) while a better agreement was found between the SVPs and

the radar (Fig. 5c, e). In the first twelve hours from the release, the wind blew quite consistently in the west-southwest direction (Fig. 5a) while the current directions were north to northwest (Fig. 5b-d). The lack of drogue in the CARTHE drifter makes it more sensitive to the Stokes drift (Novelli et al., 2017), thus, resulting in reversal of the CARTHEs (Fig. 5c, e) compared to that of the CARTHE drifter vectors (Fig. 5b, d). Even showing a high correlation ($\alpha \sim 0.81$ - 0.86, Table 4), the slip velocity caused by the wind may account for the observed discrepancy of approximately $10\,\mathrm{cm\,s^{-1}}$ in the downwind direction between

the radar and the undrogued CARTHE drifters, i.e., during 0h to 06h, 10 November 2020 and 06h to 18h, 11 November 2020 (Fig. 5b, d).





The 2dVar method performed slightly better than the traditional LS method in all common statistics while increasing the data coverage, with a complex correlation $\alpha$ about 0.77 to 0.90, an RMSE about 7 - 9 cm s$^{-1}$ (Table 4). The positive bias speed was found across nearly all types of drifters except for the nearshore group in 2020 deployment, which ranges from 2 to 4 cm

s$^{-1}$ with the SVPs and 7 to 8 cm s$^{-1}$ with the CARTHEs. This can be explained by the underestimation of the radar-derived velocities by approximately 2 cm s$^{-1}$ due to the smoothing effects of spatial ($\sim$ 13 km) and temporal averaging ($\sim$ 1 hour) which was similar to the study of Rypina et al. (2014). The $\theta$ value representing the rotation of the drifter and the radar vectors showed a larger spreading angle up to 5° between both methods (Table 4). This was more apparent for the drifter at the edge of the radar domain, such as for the near-shore group drifter in 2020, perhaps it was caused by the constrain of the 2dVar algorithm

to maintain the consistency of the velocity map Yaremchuk et al. (2016). Other than that, a more variable $\theta$ value was found across all the drifters (Table 4), possibly due to a strong vertical shear between radar velocity and different types of drifter velocity. The included radar velocities extrapolated by the 2dVar method slightly reduced the correlation $\alpha$ from 0.85 to 0.77, however, the RMSE increased to about 9 to 12 in the 2020 experiment and 12 to 15 cm s$^{-1}$ in the 2023 experiment. Rather than the increase of noise level in the radar measurements, perhaps the main reason for the increase of uncertainties in the 2023

experiment was mainly due to the extra interpolation necessitated by the large decrease in coverage of radial measurements, which was often lower than 80% ($\xi = 0.22$) and occasionally dropped to only about 10% of the radar domain (not shown).

### 4.2   Spectral analysis of the EAC jet and shelf velocities

For a better understanding of the surface circulation variability, we performed a spectral analysis of the surface current velocity time series. Three sets of approximately 2-years velocity time series were extracted at two locations (one co-located with the

CH100 mooring point in the COF region near the EAC core, and the other located offshore of the NEWC region) (152.53°E, 33.00°S). The core velocity of the EAC is identified from the jet-following method as from Archer et al. (2017b). The length of the analysis period is selected to maximize the overlapping operating period between the two radar systems while maintaining an acceptable data length. Fig. 6 represents the velocity variance spectrum computed from the extracted surface velocity in the COF and NEWC region.

Spectral analysis of the two datasets indicates similar patterns. The slope fittings across all scale bands for the core velocity, CH100, COF, and NEWC are approximately -1.54, -1.60, -1.59, and -1.55 respectively, which are quite similar to the energy cascading theory of the Taylor scaling (*-5/3=-1.67*) in the Kolmogorov spectrum (Fig. 6). Spectra derived from the mooring and radar velocity data (Fig. 6b) demonstrate that the EAC jet varies over multiple timescales - from short tidal and weather periods up to months long. Weather and small-scale disturbances can trigger the EAC jet variability at the synoptic timescale,

which is a relatively short period of 3-20 days. It has been shown that the EAC exhibits intrinsic variability driven by wind stress variations. Regional wind stress variations with periods shorter than 56 days enhance the EAC extension mean transport (Bull et al., 2017). Longer period variations also occur, with cycles around 65-100 days and 25-40 days (Fig. 6a). These variability timescales are linked to the intrinsic variability of the jet meandering on and off the continental shelf region, as documented in previous studies (e.g., Bowen et al. (2005); Sloyan et al. (2016); Archer et al. (2017a)).

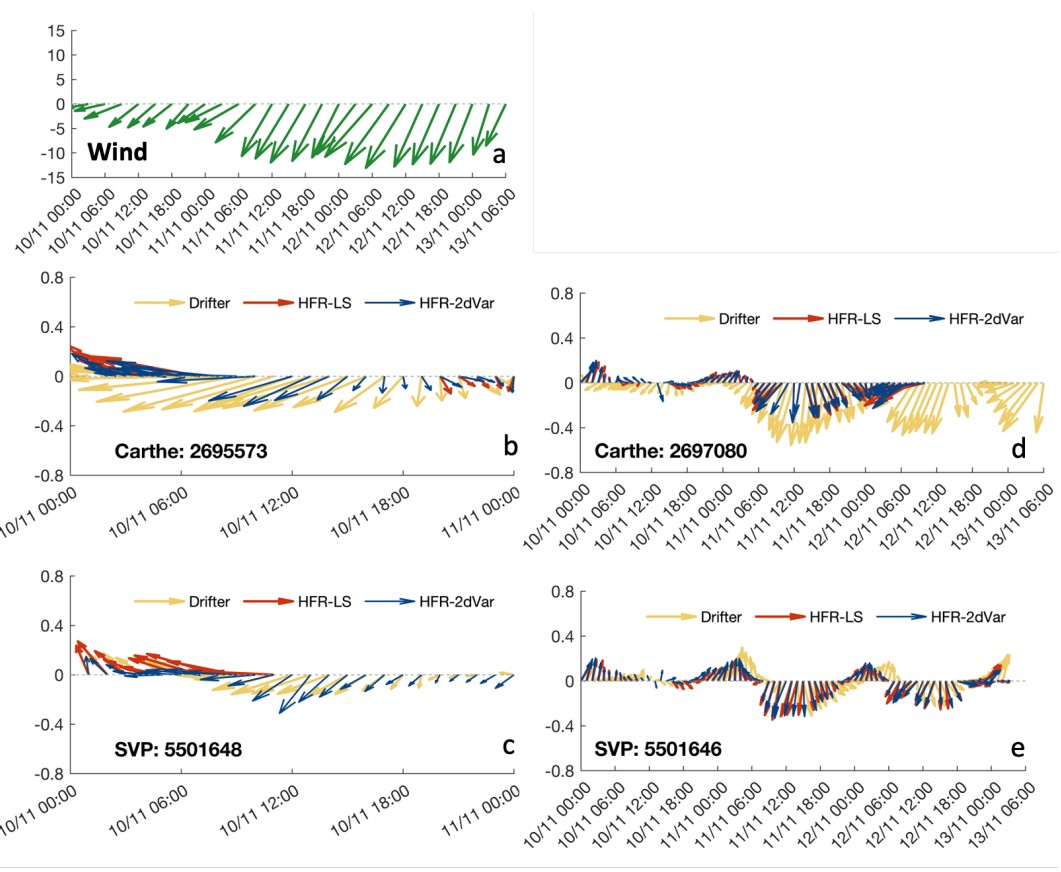

**Figure 5.** An example of wind averaged over the NEWC radar region, for the drifter deployment from November 10, 2020 to November 13, 2020, is presented (a). The velocities of drifters deployed close to the shore (b, c) and in the middle region (d, e) are shown, superimposed by the radar-derived velocities extracted along the drifter trajectories using two reprocessing methods. The unit is in m s$^{-1}$.

Since the EAC is an important driver of circulation in the region, both upstream and downstream shelf regions exhibit similar timescales though the low-frequency peaks are shifted due to the modulation of the shallow waters. Distinct spectral peaks are observed at tidal frequencies, monthly to bi-monthly (25 - 60 days) and within the range of mesoscale bands from 70 to 200 days (Kerry and Roughan, 2020). The intra-annual peaks (> 100 days) in both upstream and downstream are more pronounced in the shelf velocities (Fig. 6b, c) compared to the EAC jet velocities (Fig. 6a) and also present in the spectra derived from mooring data. The spectrum from the COF radar data indicates the energy within the bands from 25 to 40 days is comparable to that of the intra-annual bands and is roughly half an order of magnitude greater than in the downstream at similar frequencies (Fig. 6b). The energy within these bands is broad and extends toward the synoptic bands indicating that the variability of shelf waters at these timescales is likely driven by the energetic meandering of the EAC and its characteristic periods of eddy shedding (Archer et al., 2017a; Ribbat et al., 2020).




**Table 3.** Comparison of the hourly radar-derived velocities processed with the two different gap-filling (LS or 2dVar) methods with moored velocity datasets at COF. The moorings are located above the 70 m isobath (CH070) and 100 m isobath (CH100) with currents measured at a depth of $\sim 9$ m below the surface with data used from July 2012 - October 2020. The gap-filling method (LS or 2dVar) is indicated. N is the number of data points used in the comparison, $\alpha$ is the complex correlation, $\theta$ is the phase difference in degrees, and the bias and root mean square errors (RMSE) (m s$^{-1}$) in u and v and total are shown.

| Comparison | Method | N | $\alpha$ | $\theta$ | Bias u | Bias v | RMSE u | RMSE v | Bias Total vel. | RMSE Total vel. |
|---|---|---|---|---|---|---|---|---|---|---|
| **COF** | | | | | | | | | | |
| CH070 | LS | 38583 | 0.69 | 8.7 | 0.11 | -0.01 | 0.28 | 0.15 | -0.09 | 0.22 |
| | 2dVar | 38583 | 0.73 | 8.7 | 0.09 | -0.01 | 0.25 | 0.14 | -0.08 | 0.18 |
| CH100 | LS | 34389 | 0.88 | -0.7 | 0.04 | 0.05 | 0.17 | 0.16 | -0.07 | 0.19 |
| | 2dVar | 34389 | 0.89 | -0.5 | 0.03 | 0.05 | 0.15 | 0.16 | -0.07 | 0.18 |
| **COF - 25h lowpass** | | | | | | | | | | |
| CH070 | LS | 38583 | 0.73 | 8.9 | 0.11 | -0.01 | 0.25 | 0.13 | -0.08 | 0.19 |
| | 2dVar | 38583 | 0.76 | 9.2 | 0.09 | -0.01 | 0.22 | 0.12 | -0.07 | 0.16 |
| CH100 | LS | 34389 | 0.9 | -0.7 | 0.04 | 0.06 | 0.14 | 0.14 | -0.07 | 0.16 |
| | 2dVar | 34389 | 0.92 | -0.5 | 0.04 | 0.04 | 0.13 | 0.14 | -0.07 | 0.16 |

**Note**: Phase difference $\theta > 0$ - In-situ vectors rotate counter-clockwise to the HFR vectors.

$Bias < 0$ - in-situ data is smaller than the HF radar measurement and vice versa.

On the other hand, discrepancies between the mooring and the radar-derived velocities can be found in the synoptic range (3-20 days), where the radar data exhibits higher variability, approximately half an order of magnitude greater than the mooring data (Fig. 6b). This difference likely arises from the different depths of the observations, 10 m for CH100 mooring compared to 0.9 meters for COF radar. Upon closer inspection, small and broad synoptic peaks (3-20 days) are evident in both the EAC variability and the shelf velocities though remaining weak downstream. However, these synoptic signals are not as pronounced in the mooring velocity measurements. The enhanced synoptic variability in the radar data is likely due to local motions related to weather fluctuations and frontal eddies (Schaeffer et al., 2017). These smaller-scale instabilities are typically located inshore of the jet, resulting in a stronger energy level in the spectrum than the jet offshore.

In the high-frequency ranges, all spectra indicate the same level of energy with the semi-diurnal peaks (M2 and S2) being smaller and narrower than the diurnal peaks (K1 and O1) (Fig. 6b, c). The semi-diurnal peaks in the NEWC data are even smaller than in the COF region, suggesting the dominance of the diurnal tidal regime in this region (Fig. 6c). Within the diurnal bands, the near-inertial frequencies closely match the diurnal frequencies, especially at COF (23.6 hours for COF and





**Table 4.** Comparison of the hourly radar-derived velocities processed with different methods with the drifter datasets in NEWC. The drifter velocities were taken from 2 different types of surface drifters, Carthe with a drogue depth of 60 cm and the SVP with a drogue depth of ~15 m. An additional comparison with the undrogued CARTHE drifters (representing the top few cm of the water column) is shown. The lack of the drogue makes the drifter more susceptible to surface wind. The numbers within the brackets represent comparisons made with extrapolation from using the gap-filling 2dVar method. The unit of bias and RMSE are in m s$^{-1}$. The average level of gap, $\xi$, is shown for each drifter deployment

| | | LS | | | | | 2dVar | | | |
|---|---|---|---|---|---|---|---|---|---|---|
| | N | $\alpha$ | $\theta$ | Bias | RMSE | N | $\alpha$ | $\theta$ | Bias | RMSE |
| **CARTHEs** | | | | | | | | | | |
| 2020 ($\xi = 0.03$) undrogued | | | | | | | | | | |
| - Near shore | 18 | 0.90 | -3.9 | 0.06 | 0.06 | 18 (28) | 0.86 (0.83) | 4.1 (-0.9) | 0.08 (0.11) | 0.09 (0.12) |
| - Offshore | 52 | 0.80 | -6.7 | 0.06 | 0.08 | 52 (59) | 0.81 (0.82) | -8.6 (-6.3) | 0.06 (0.06) | 0.08 (0.09) |
| 2023 ($\xi = 0.22$) | | | | | | | | | | |
| undrogued | 29 | 0.88 | 27.7 | 0.08 | 0.11 | 29 (114) | 0.90 (0.77) | 32.3 (8.1) | 0.05 (0.10) | 0.09 (0.15) |
| drogued | 33 | 0.89 | 19.4 | 0.06 | 0.12 | 33 (125) | 0.89 (0.85) | 24.7 (5.6) | 0.04 (0.09) | 0.12 (0.15) |
| | | | | | | | | | | |
| **SVPs** | | | | | | | | | | |
| 2020 ($\xi = 0.03$) | | | | | | | | | | |
| - Near shore | 37 | 0.87 | 6.1 | -0.07 | 0.07 | 37 (98) | 0.88 (0.80) | 11.7 (17.8) | -0.02 (-0.05) | 0.07 (0.08) |
| - Offshore | 199 | 0.81 | -3.0 | 0.04 | 0.09 | 199 (222) | 0.83 (0.83) | 2.0 (1.8) | 0.04 (0.03) | 0.08 (0.08) |
| 2023 ($\xi = 0.22$) | | | | | | | | | | |
| | 110 | 0.79 | 16.6 | 0.04 | 0.15 | 110 (306) | 0.77 (0.78) | 17.7 (12.4) | 0.04 (0.08) | 0.15 (0.16) |

**Note**: Phase difference $\theta > 0$: Current meter velocities rotate counter-clockwise to the HF radar vectors.

$Bias < 0$: Current meter velocities are smaller than the HF radar measurements and vice versa.

22.0 hours for NEWC). As a result, the diurnal energy can be affected by the inertial motions. The strong diurnal variability can also result from near-inertial period motions (at around 30°N and 30°S) close to the diurnal forcing driven by tides and land-sea breezes. This also can be influenced by the background vorticity or potentially amplified at the critical latitudes (Archer et al.,
2017a).

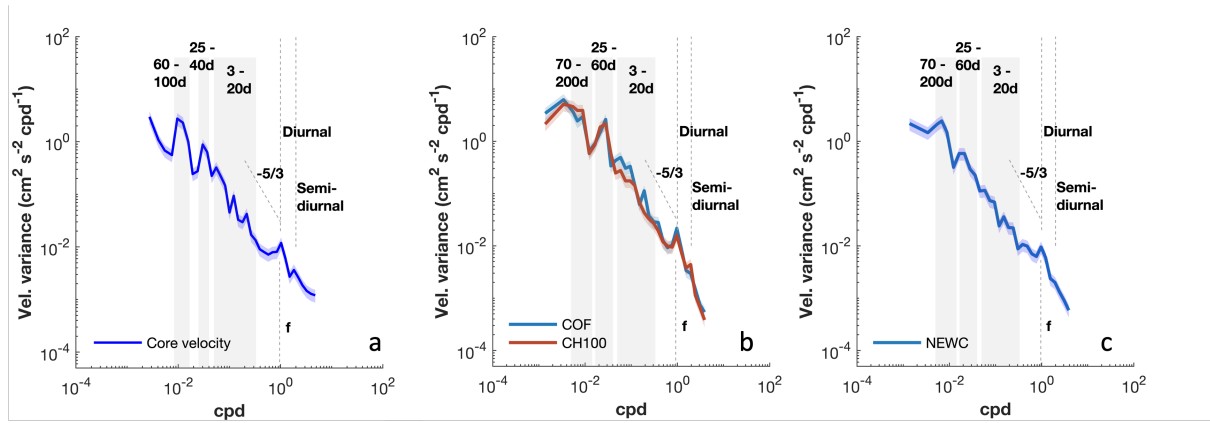

**Figure 6.** Spectral analysis of the hourly radar-derived current speeds during a 2 year period (January 01, 2019, to January 01, 2021) of the EAC core velocity (a), a point at the CH100 mooring location (153.39°E, 30.26°S) in COF (b) and one above the 200m isobath off NEWC (152.53°E, 33.00°S) during September 01, 2019, to September 01, 2021 (c). Inertial frequencies (*f*) of 23.6 h and 22.1 h in COF and NEWC respectively are shown in the spectrum. The slope *-5/3* represents the theory of energy cascading in the inertial range of the Kolmogorov turbulent spectrum. The 95% uncertainties are shown by the blue and red shading.

## 4.3 Tidal variability

Hourly radar measurements can be used to examine tidal dynamics in the region. To further examine the capability of observing tidal currents from HF radar, the major tidal constituents (M2, S2, K1, and O1) are estimated from hourly COF radar and mooring data using the UTide toolbox (Codiga, 2011) during the year 2019, suggesting a general agreement in the shape and
size of the tidal ellipses (Fig. A4).

The tidal harmonic analysis is shown in Fig. 7. The spatial distribution of total velocity variance shows that tidal motions contribute a minor portion to surface current variability in both regions. This contribution is approximately 5% in the off-shore region of COF but is higher towards the shallow shelf in NEWC, with a contribution of around 15%. The average tidal contribution to total surface variability is generally less than 10%, as the shelf circulation is predominantly influenced by the
EAC.

Regarding tidal magnitude, the upstream region's tidal regime is primarily governed by semi-diurnal tides. The M2 tidal magnitude (about 0.04 m s$^{-1}$) is twice as large as the diurnal tides (K1 and O1, approximately 0.02 m s$^{-1}$), as shown in Fig. 7. Closer to the shore downstream, the diurnal tidal regime becomes more significant. The K1 tidal magnitude, derived from NEWC radar, is the most substantial among the four tidal constituents, with a value of roughly 0.06 m s$^{-1}$ (Fig. 7).


Nonetheless, the overall tidal magnitude remains small, varying from 0.02 m s$^{-1}$ offshore to about 0.06 m s$^{-1}$ in the shallow

waters.

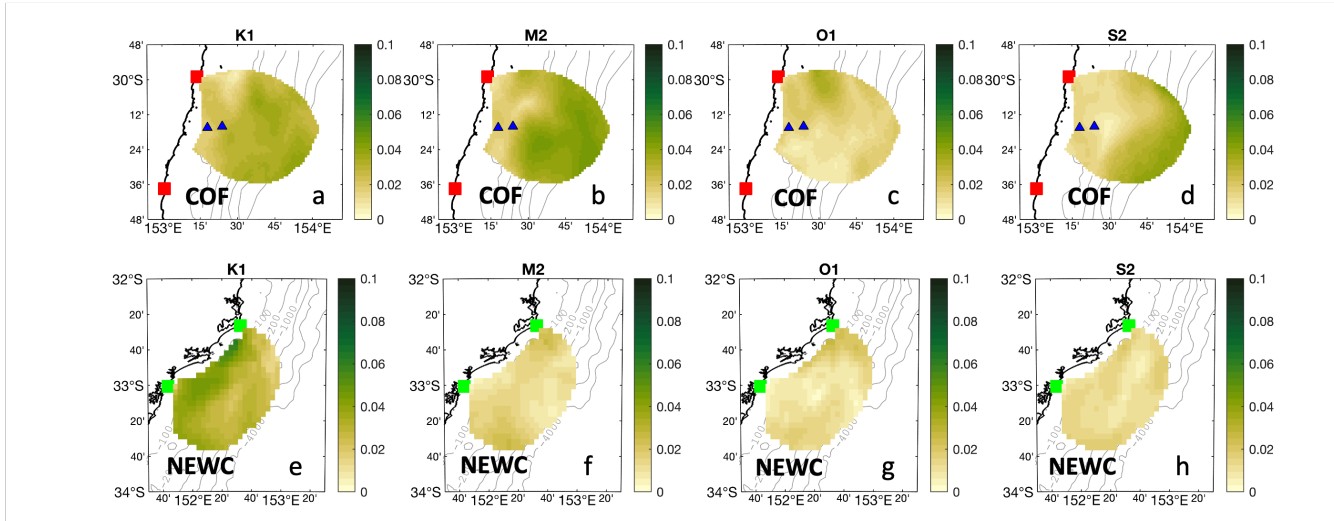

**Figure 7.** Magnitude of the major tidal ellipses derived from the 1-year period (01 January 2019 to 01 January 2020) radar current velocities for four main tidal constituents, diurnal (K1 and O1) and semi-diurnal tides (M2 and S2).

## 4.4 Seasonal and spatial variability of the surface currents

Next we examine how the shelf circulation responds to the seasonal cycle of the EAC. We begin our analysis by examining the current velocities upstream of the separation zone using COF radar data. To calculate the monthly mean surface currents

at COF, we use an average of the data grid cells that cover a period of more than five years. Throughout the year, the monthly mean current vectors consistently point poleward, with notable variation in their magnitude (Fig. 8). The radar-derived current patterns indicate the EAC strongly modulates the local circulation, with mean current velocities gradually increasing offshore. The oceanic jet follows the isobaths and is influenced by the underlying topography. This is shown by the jet boundary, visually identified by the velocity line of -0.6 m s$^{-1}$, which rarely extends inshore beyond the 200 m isobath (Fig. 8). The core of the

East Australian Current (EAC), indicated by the highest mean velocities, flows in a region above the 1500 and 2000 m depth contours, which is consistent with previous findings by Archer et al. (2018). The jet funneling effect, as detailed by Oke and Middleton (2000), is apparent in the southern domain. The term refers to the narrowing and increasing of the jet intensity through the narrow shelf. It was shown by the higher mean velocity and the current variance ellipses that orient towards the shore at the south of the domain (Fig. 8). Corresponding to the variability of the EAC, the strongest monthly averaged currents

are observed during the austral summer and weaker during the austral winter. In November and February, the EAC strengthens, and the radar-derived shows the highest velocities reach approximately 1.4 m s$^{-1}$ at the core of the EAC with high variation





(approximately 0.3 m s$^{-1}$). However, during June, the EAC footprint is minimal, resulting in much weaker mean currents (Fig. 8).

Compared to the upstream region, the NEWC data is more limited as the data spans only 4 years. January and December are the two months with the lowest data available of about 2 years (Fig. A3). Despite this limitation, it is evident that the surface currents quickly lose momentum and decreases in magnitude at the separation region (Fig. 9) compared to upstream. This results from the fact that the EAC typically separates around this latitude, bifurcating into the eastern and southward extensions (Oke et al., 2019). Past the separation point, the southward extension of the EAC broadens and shallows, becoming more barotropic and decreasing its poleward transport (Kerry and Roughan, 2020). Around 32.5°S, the EAC signature from the mean velocity field appears to turn eastward and veer away from the continental shelf, likely experiencing an "inertial overshooting" effect from the westward bending shelf (Oke and Middleton, 2000). Based on the radar-derived data, we found that the current evolution downstream does not entirely follow the seasonality of the currents upstream and demonstrates a more complex variability. This can be shown by the maxima of current velocity flowing along the shelf that occurs during February ( 0.6 m s$^{-1}$) while during March and April, the strong current patterns persist and shift more offshore (Fig. 9). On the other hand, the downstream average current speed decreases during November and December off NEWC, (but increases at COF Fig. 8). Given the differences between the monthly variability upstream and downstream, one may suggest that it is due to the difference in the average period of both data sources. However, it has been shown that the circulation downstream of the EAC is rather complex and closely related to the mesoscale circulation driven by the EAC eddy-shedding timescales (Kerry and Roughan, 2020).

The intra-annual change of the surface circulation was examined using the monthly averaged velocity (Fig. 10. The daily radar-derived velocities in both regions were averaged domain-wide to assess the mean and standard deviation for each month of the year. Velocities were rotated by 18 and 30 degrees for COF and NEWC data, respectively, to derive the along and cross-shore components, based on the predominant orientation of current variance ellipses (Fig. 8 and 9). The monthly averaged velocities indicate a poleward mean transport (negative along-shore velocities) at both sites consistent throughout the year (Fig. 10). The COF radar-derived velocities show that the annual cycle of poleward transport exhibits peaks in the early and late summer (November and February) at around 0.80 m s$^{-1}$, and the lowest occurs in the winter months (June and July) at about 0.25 m s$^{-1}$. The range of monthly variation, approximately 0.4 m s$^{-1}$, is larger than the standard deviation and exceeds the average wintertime velocity. (Fig. 10b). The monthly patterns reveal slight increases in along-shore velocity during autumn (April) and spring (August to October).

Off NEWC there is a strong annual cycle in the along shore direction, although the velocities are weaker, ranging 0.4 - 0.1 m s$^{-1}$ from summer to winter respectively (Fig. 10b). On the other hand, the cross-shore component in both regions does not show a seasonal pattern but varies significantly throughout the year. The increase in cross-shelf exchange upstream is driven by the meandering of the surface-intensified jet on and off the shelf (Malan et al., 2022), thus, the variability occurs both in magnitude and sign. The mean cross-shelf velocity downstream remains similar, however, the standard deviation is half as much as that found upstream (Fig. 10a).

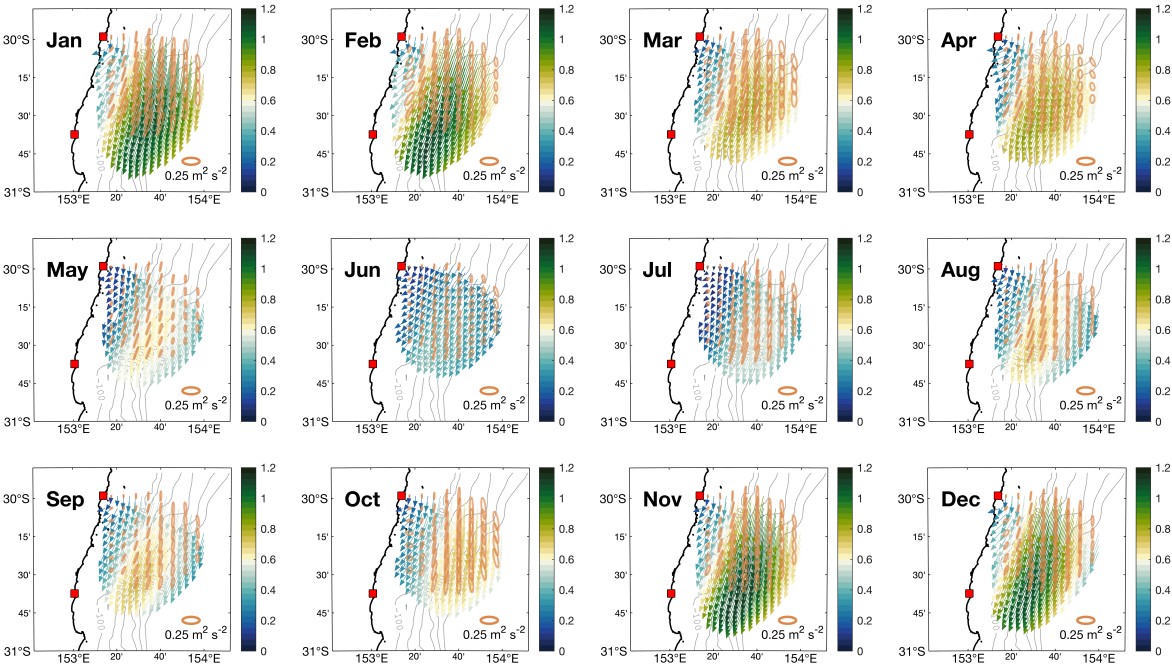

**Figure 8.** Maps showing the monthly mean radar-derived current vectors at Coffs Harbour (upstream) using hourly data from the COF radar from March 2012 to February 2021. The velocity unit is m s$^{-1}$. The current velocity variances are illustrated by plotting ellipses at 6-grid point intervals for visualization. The bathymetry contours are plotted at 100, 200, 1000, 2000, 3000 and 4000 m levels.

### 4.5 Seasonal and intra-annual variability of the EAC jet

The EAC proper demonstrates a pronounced seasonal cycle in temperature and velocity, a subject that has been explored in previous studies (e.g., Godfrey et al. (1980); Ridgway and Godfrey (1997)). Moreover, Archer et al. (2017a), using the jet-following method and 4-year radar data from COF, revealed that the EAC magnitude and its associated variance follow a

seasonal pattern, peaking during summer. Here we extend the analysis and summarize the key findings of the EAC characteristics using a longer dataset of around 8 years. As the EAC consistently oscillates on and off the continental shelf, with the offshore extension of the jet core up to 90 km from the coastline (Archer et al., 2017a) and disappearing from radar footprint, the jet-following method, used in Fig. 11, enables the detection and characterization of EAC signals within its coherent structure. The jet exhibits an asymmetric behavior. Indeed, on its eastern side, it can freely meander and shift laterally over the

open ocean, but its western flank is constrained by the presence of the continental landmass limiting how far it can veer in that direction. What can be seen from this analysis is the monthly fluctuation in the EAC's intensity, peaking during mid-summer



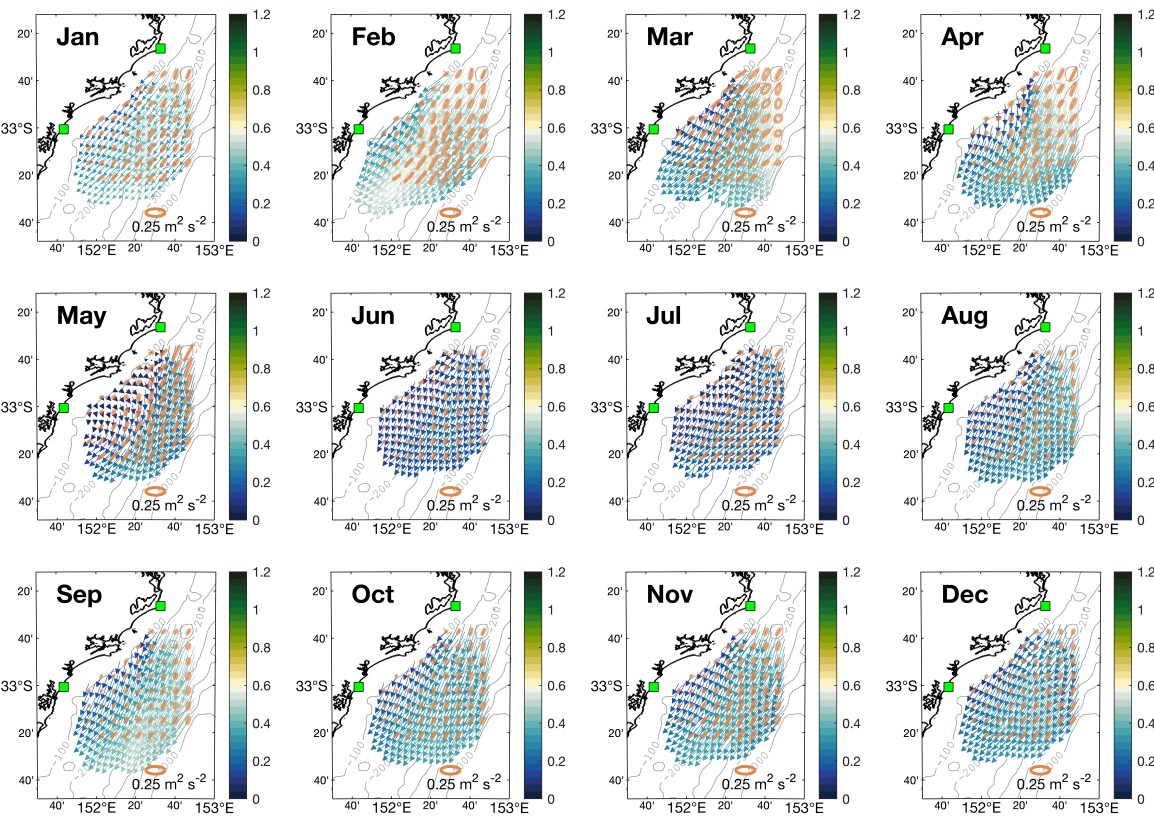

**Figure 9.** Maps showing the monthly mean radar-derived current vectors off Newcastle using hourly data from the NEWC radar from November 2017 to February 2024. The velocity unit is m s$^{-1}$. The current velocity variances are illustrated by plotting ellipses at 2-grid point intervals for visualization. The bathymetry contours are plotted at 100, 200, 1000, 2000, 3000 and 4000 m levels.

(January) (mean speed approx. 1.5 m s$^{-1}$) and reaching its lowest in winter (June) (mean speed approx. 0.9 m s$^{-1}$), with the annual difference (0.6 m s$^{-1}$) of around 40% of the mean velocity (1.35 m s$^{-1}$) (Fig. 11). The jet-following method identifies a stronger EAC in late winter (July and August), with velocities around 1.2 $\pm$ 0.2 m s$^{-1}$), however, contrasting with the weaker

jet speeds of approximately 0.6 $\pm$ 0.2 m s$^{-1}$) observed in the monthly mean surface velocity map (Fig. 8). The EAC jet exhibits from May to August large variability during winter and this variability is more pronounced on the right flank than on the left, as the jet is narrower in winter (Fig. A2). As shown by the extension of the jet cross-structure, the jet can be widened during December to February when the EAC is strongest (approximately 60 km) but also the widening jet can be found during June when the EAC is weakest. The variability of the jet width over the year exhibits a range of variation around 10 km (Fig. 11).

Earth System Science Data Discussions — Open Access

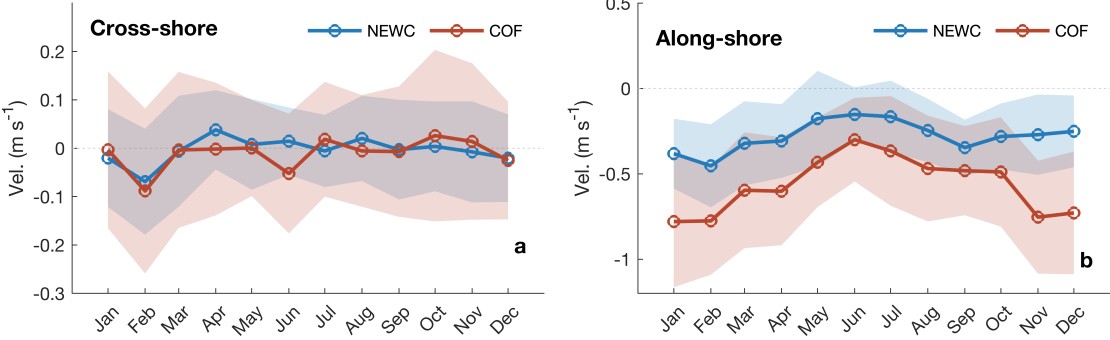

**Figure 10.** Domain-averaged monthly radar-derived velocity plotted as a function of the month for COF (a) and NEW (b), respectively. The velocity vectors are rotated 18 degrees clockwise for COF and 30 degrees clockwise for NEWC to obtain the cross-shore and along-shore velocities accordingly. A positive value of the along (cross)-shore velocity represents the northward (off-shore) transport and vice versa. The shaded area denotes the standard deviation of domain-averaged velocities.

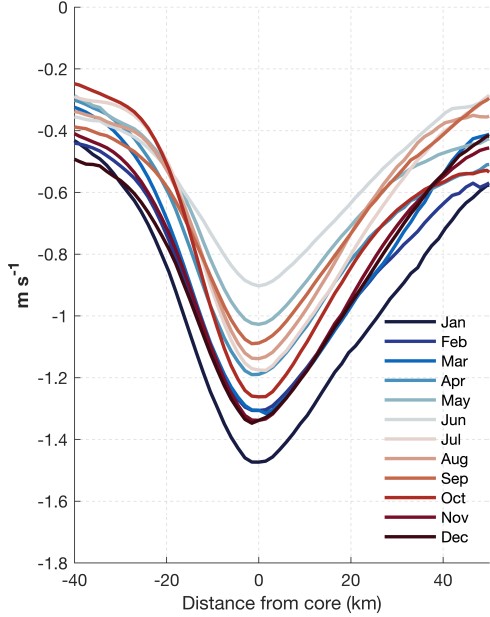

**Figure 11.** Annual cycle of the horizontal structure of the EAC jet as identified using the jet-following method of Archer et al. (2017b) based on 8 years of Coffs Habour (COF) data. Negative values indicate the poleward velocity of the EAC. Velocities are averaged in a band from $30.2°$ to $30.6°$S.





## 4.6 Inter-annual variability of the EAC as observed from the radar

The poleward transport anomalies of the EAC in the COF radar data also show strong inter-annual variability (Fig. 12c). The lowest transport occurred in late June - November 2017, which is consistent with the observations of a large and long-lived cyclonic eddy generated during that time of the year (Roughan et al., 2017). The data also suggest a shift in the jet core position from late 2019 to mid-2020, as demonstrated by the maximum core velocity moving toward the 1000m isobath, which led to a significant increase in the averaged shelf velocity (Fig. 12a). This shift was followed by the reversal to its average position by late 2020 (Fig. 12c). A similar event appears to have occurred in January 2015, marked by an increase in along-shelf velocity as shown by both mooring and radar data (Fig. 12c).

The velocity time series from the CH100 mooring and data extracted at the same location from COF radar are shown for comparison (Fig. 12c). The estimated inter-annual variation contribution in total along-shore velocity variances from COF and CH100 mooring observation were nearly identical, about 12% and 10%, using the radar-derived velocity and CH100 mooring data, respectively. Results from the COF along-shore velocity and the mooring velocities show a significant change in the timescale of more than 4 years which is shown by two phases of the increasing and decreasing of the EAC core velocity (Fig. 12a. The EAC signals observed from the CH100 mooring location became stronger during January 2015 with the velocity anomaly exceeding 0.25 m s$^{-1}$ (0.3 m s$^{-1}$ from the CH100 mooring) while it weakened during the following summer (Fig. 12c). From 2017 onward, another pattern of strengthening and weakening of the EAC is consistent with the mooring observations. Another weak period of the EAC can be seen from the CH100 mooring occurred from January 2020 and lasted until January 2023, however, unfortunately, the COF radar data were not available for this whole period (Fig. 12c). Despite the differences between the two datasets, the consensus on the EAC patterns reinforces our method's reliability for processing radar measurements over inter-annual and longer timescales.

Downstream of the EAC, the inter-annual variability accounts for about 14% of total variability for both along and cross-shore velocity (not shown). Although the mean alongshore velocity downstream is approximately half of the upstream mean velocity, it surprisingly exhibits inter-annual variations of up to 0.4 m$^{-1}$, comparable to the more energetic environment upstream (Fig. 12a). The interannual variability of the NEWC along-shore velocity exhibits a complex pattern. Maximum poleward transport through the NEWC region occurred from January to May 2022, coinciding with the weakened mean EAC intensity indicated by CH100 mooring data. Conversely, the southward velocities anomaly decreased in 2023 as the EAC intensified (Fig. 12c).

## 5 Discussion

### 5.1 The gap-filled HF radar total surface currents

Obtaining continuous, gap-free surface current measurements from HF radar systems is challenging due to factors like radio interference and environmental noise. The advanced gap-filling approach, two-dimensional variational data assimilation (2dVar), was used to reprocess the surface current velocities from HF radar measurements. A rigorous comparison of radar observations

Earth System
Science
Data

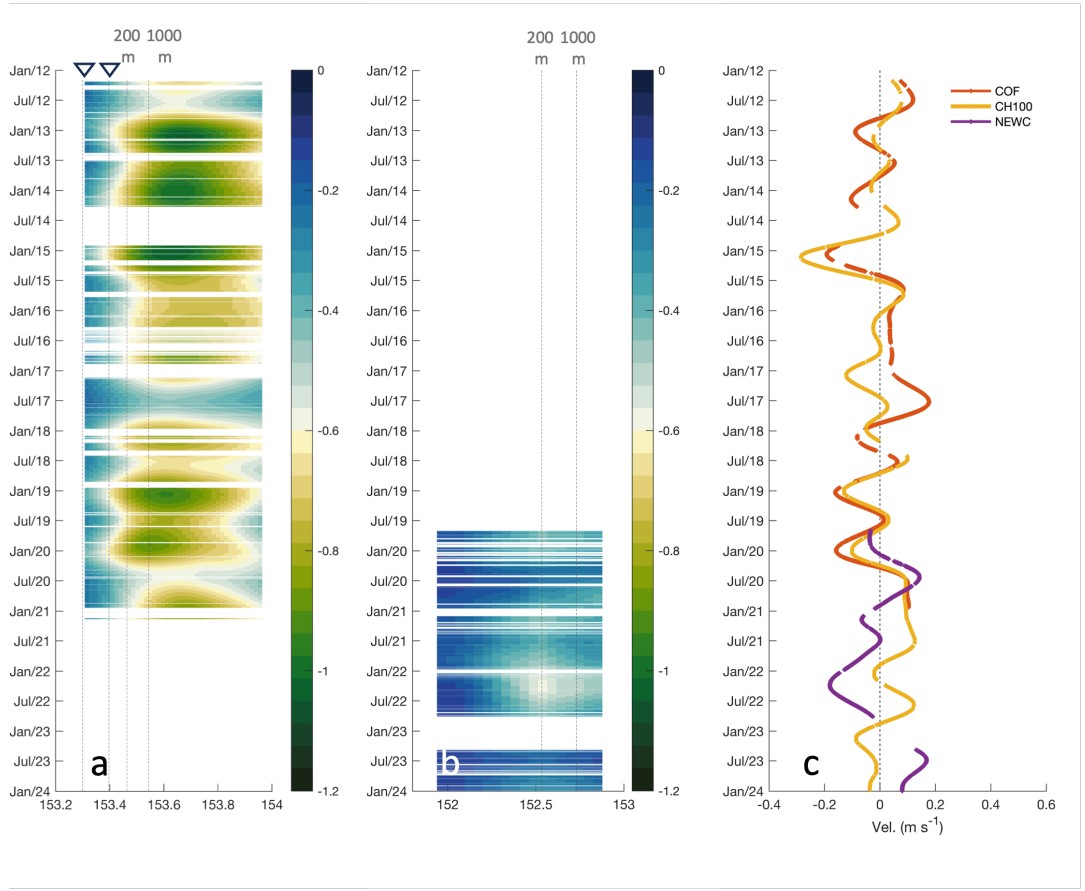

**Figure 12.** Hovmoller plot showing inter and intra annual variability of the 1-year low-pass filtered radar-derived along-shore velocity at Coffs Harbour (COF) in m s$^{-1}$ along a line normal to the shore from $30.26^o$ S (a) and Newcastle (NEWC) radar along the line $33.06^o$ S (b). The velocity vectors are rotated 18 degrees clockwise for COF and 30 degrees clockwise for NEWC to obtain the along-shore velocities. The triangles denote the CH070 and CH100 mooring locations, respectively. The dashed lines represent the 70m, 100m, 200m, and 1000m isobaths in (a) and the 200m and 1000m isobaths in (b). Also shown are the along-shore velocity anomaly derived from COF radar extracted at the CH100 mooring location (red), from mooring data (orange), and NEWC radar (purple) extracted at the same point as from the spectral analysis (152.53°E, 33.00°S). A negative (positive) velocity indicates the current's southward (northward) movement. All time series are smoothed by 1-year Butterworth low-pass filtering.

in two distinct regions of study was done to assess the performance of the gap-filling method. Several studies, such as, Molcard et al. (2009) found the RMSE ranges from 6 to 10 cm s$^{-1}$ between 45 MHz radar and the Coastal Dynamics Experiment (CODE)-style (drifting at $\sim$ 1 m), Kirincich et al. (2019) compared the 25 MHz HF radar in the Martha's Vineyard with the CODE drifter found the RMSE within the range of cm s$^{-1}$ and correlation of around 0.73. Kalampokis et al. (2016) found an RMSE of 10 cm s$^{-1}$ and a correlation coefficient of 0.8-0.85. In another study, Capodici et al. (2019) found the RMSE ranges from 8 to 18 cm s$^{-1}$ for the 9 MHz radar with the SVP drifters. Thus, it suggests the results of our study are encouraging.





The comparison revealed that the velocities processed using the 2dVar approach demonstrated a slightly better performance compared to the LS method (typically applied to IMOS data). The improvement of performance shown by the comparison with the CH070 mooring data and the trajectories of the surface drifters (Table 3 and Table 4), which mostly remained within the offshore limits of the radar domain, suggests that this improvement is probably due to the smoothing of outliers at the periphery of the radar domain, as evidenced in Fig. 3. The 2dVar approach incorporates additional constraints and spatial smoothing, which reduces the impact of erroneous or spurious measurements, particularly effective with those occurring at the edges of the radar coverage or within the area where the radials are limited for the LS to reconstruct the total velocity properly. To that extent, the 2dVar algorithm can reconstruct a more coherent and consistent representation of the flow dynamics, thereby enhancing the overall accuracy and reliability of the derived velocity estimates (Yaremchuk and Sentchev, 2011). Furthermore, gap-free current fields are especially important for applications like Lagrangian tracking of pollutants or debris, where gaps can lead to inaccurate trajectory calculations.

Typically, the very surface ocean layer (representing the top few centimeters and meters from the surface) is highly variable and related to the underlying ocean velocity, atmospheric forcing, boundary turbulent processes, and wave dynamics. The understanding of this very surface dynamics would benefit the improved accuracy of determining the transport of material (e.g., oil slicks and buoyant organisms) at the ocean surface or, to some extent, influencing the air-sea momentum and heat fluxes (e.g., Janssen and Viterbo (1996); Shimura et al. (2017, 2020)). To assess this theory, the NEWC HF radar measurements, representing a depth of 2.3 meters, were evaluated against data from undrogued drifters (sampling the top 0-5 cm of the water column), drogued drifters (sampling 0-60 cm), and SVP drifters (sampling ~15 m). This multi-instrument approach allowed us to assess the HF radar performance across various depth ranges and investigate its ability to capture surface and near-surface current dynamics in the absence of long-term mooring data.

The CARTHE drifter velocity measurement characteristics were shown to be very similar to the radar-derived velocity, however, were substantially stronger than that recorded by the radar on average (Table 4). The large difference in RMSE may be due to the HF radar effectively measuring currents at a depth approximately two meters deeper than the CARTHE drifter and derived from the fact that the radar-derived velocities were averaged over 13 km and 1 hour. Notably, the CARTHE drifter behavior in the first few hours on 11 November 2020 indicated a large portion of the surface and subsurface difference even during the low wind condition (wind speed was lower than 10 m s$^{-1}$) (Fig. 5a). Despite the high correlation, the bias and RMSE observed between the radar-derived velocities and the CARTHE drifter were surprisingly large compared to that with the SVP drifter, particularly for the drogued CARTHE which was designed to minimize windage and Stoke drift. The discrepancies may arise from the difference in effective measurement depths, approximately 2.3 meters for the 5.3 MHz long-range radar and 0 to 60 centimeters for the CARTHE drifter, leading to the notable bias between the dynamics of the very-near surface and the lower layers.

A strong shear was found for CARTHE drifters close to the shore and lasted for a few hours. The mechanism for the shear between the very near surface and the radar depth remains unknown. During the deployment in 2020, the undrogued CARTHE drifters showed a discrepancy between the drifter and the radar-derived velocity of approximately 10 cm s$^{-1}$ in the downwind direction drifters. The CARTHE drifter velocities were approximately 3 - 4% of the typical wind velocity, which





was higher than the slip velocity of 2% in the laboratory for the undogued CARTHE drifter (Novelli et al., 2017). A comparison between the HF radar and the undrogued CARTHE in Western Australia, as discussed in van der Mheen et al. (2020), suggests

incorporating a 3% drift factor into the Lagrangian model to more accurately represent the near-surface transport using the HF radar. However, a more careful study with a higher number of similar drifters (about 7 in this study) in similar conditions should be considered in the near future.

## 5.2 The EAC variability

Analysis of surface currents from the continuous gap-filled HF radar current maps over the EAC reveals distinct annual cycles

of the surface circulation over the Eastern Australian Shelf. The along-shelf flow, largely influenced by the poleward-flowing East Australian Current, exhibits a pronounced seasonal signal with maximum southward velocities during the austral summer (December-February) and a weakened flow in winter (June-August). This seasonality is primarily modulated by the annual strengthening and relaxation cycle of the EAC western boundary current dynamics. In contrast, the cross-shelf circulation displays a rather complex annual cycle with substantial variations across both the upstream and downstream regions. This

complex cross-shelf flow is primarily governed by the meandering behavior of the EAC while the interplay between wind forcing, alongshore pressure gradients, and shelf-slope processes such as upwelling and eddy shedding from the EAC also contribute significantly to the cross-shelf flow variability. The ability to resolve these annual cycles in the along-shelf and cross-shelf flows significantly contributes to the study of nutrients, larval dispersal and biological productivity across the East Australian shelf ecosystem.

Although the mean surface current velocity observed by the NEWC (Newcastle) HF radar site is relatively small, only about half the magnitude compared to measurements from the COF (Coffs Harbour) site further north, the interannual variability in the currents is quite comparable between the two locations (roughly 14% of the total variability). Despite the weaker mean flow off Newcastle, the standard deviation of the current velocities is similar to that seen in the stronger EAC-influenced currents near COF region (Fig. 12c). This suggests that while the time-averaged currents are substantially different, the range

of variability and energetic departures from the mean state are of similar magnitude at both sites. The high variability at NEWC implies the region experiences significant current variability and energy inputs (Kerry and Roughan, 2020; Malan et al., 2022). The consistency in variability levels highlights the dynamic nature of the East Australian shelf circulation, even in lower mean flow regimes. Resolving this variability is crucial for applications like particle tracking and dispersal modeling all along the shelf waters.

In addition, the 8-year COF radar data reveals an interannual variability in the core position of the EAC from 2018 to 2020 (Fig. 12a). While the EAC core continuously meanders back and forth across the shelf, this interannual variability is related to the year-to-year fluctuations in the mean location and pathway of the EAC energetic poleward flow along the continental slope. Such changes in the EAC core position are influenced by an array of factors including interactions with large-scale circulation, long-term variation, and the atmospheric circulation systems (Bowen et al., 2005; Li et al., 2023). While the specific drivers

are not explored further here, observing and understanding this interannual variability is crucial for monitoring and predicting the EAC behavior over longer multi-year time scales.



### 5.3 Limitations and recommendations

While HF radar systems provide valuable measurements of surface currents over large spatial domains, the radar datasets have certain inherent limitations. One major limitation is the spatial and temporal gaps that occur in the data due to factors like radio interference, environmental noise, or radial beam obstructions. The limitation comes from the fact that even the 2dVar approach can resolve the spatial gaps, the temporal gaps remain, such as the gaps due to hardware failure as shown by the blank space in Fig. 2. These gaps can be problematic for applications that require continuous current fields. The large discontinuity in the radar data, COF (2014, 2021-2023), and NEWC (July 2018 - July 2019), also prohibits the co-analysis between upstream and downstream.

The NEWC long-range radar footprint in theory is able to measure up to 250 km from the shore. However, the current limitation in operating transmitted power does not allow for observing the region of more than 150 km from the shore. Moreover, in recent years the offshore spatial extent has reduced compared to the start of the time series (Fig. 2) due to hardware defects producing noise and therefore limiting data range. The analysis of various drifters at the NEWC radar in 2020 and 2023 showed an increase in RMSE discrepancies between radar-derived velocity and drift velocity, rising from 0.08 - 0.11 m s$^{-1}$ to approximately 0.15 - 0.16 m s$^{-1}$ which is shown in both LS and 2dVar methods (Table 4). This larger RMSE discrepancy might be attributed to the fact that the drifters moved quickly toward the offshore region of the radar domain, which is associated with higher errors (Fig. 4). Considering the decrease in radar coverage in this period (Fig. 2), the error might be linked to the contamination by instrumental noise that affects the interpretation of radar data (Forget, 2015). It is worth noting that spikes had a more significant impact on radar observations, thus, it is also important to implement a more robust QC threshold to obtain a more precise dataset. Furthermore, the shortage in radar spatial resolution due to their relatively coarse measurement footprint, especially the NEWC radar with the reduced bandwidth acquired only 13 km spatial resolution (Table 1), likely underestimates the effect of submesoscale features ($O \sim 1 - 10$ km). Such limitations reduce the capability to acquire a full picture of the EAC jet downstream.

Overall, our current limitations include the restricted spatial coverage and the potential for data inconsistencies between the two different HF radar systems employed. We have deployed two different HF radar systems, WERA (COF) and CODAR (NEWC) due to the availability of suitable land for deployment. Following an extensive comparative analysis using data from both Lagrangian drifters and stationary moorings, we have demonstrated that both HF radar systems perform well and are suitable for research requirements. However, the WERA system appears to deliver data with notably higher consistency across both temporal and spatial dimensions (Fig. 2). Consistency in data observations is critical for research and operational forecasting systems.

To enhance the effectiveness of the EAC coastal radar system, a key recommendation emerges from our research regarding the spatial gap of the coastal radar network along eastern Australia. The large gap between our two radar sites limits the ability to fully observe the EAC behavior comprehensively along the coast. Extending our HF radar coverage in the EAC separation region, particularly in the region known as the "eddy avenue" of the East Australian Current (EAC), is crucial. This area is not only characterized by complex dynamics but also holds significant economic importance, i.e., Sydney, Australia's largest





city and a major coastal economic center. It is known that the region is currently experiencing the effect of a warming climate. The intense poleward extension of the East Australian Current (EAC) brings more equatorial warm waters poleward (Li et al., 2022), however, the link with a warming of shelf waters is unknown (Malan et al., 2021). Moreover, integrating the extended HF radar data with advanced numerical modeling techniques would enhance the accuracy and reliability of our coastal regional

forecasts. This improvement has been demonstrated by Kerry et al. (2018), highlighting the importance of a consistent data stream integrated into advanced numerical modeling frameworks.

## 6   Summary

We have produced a gap-filled HF radar data set from all available HF radar surface current datasets derived from two HF radar systems. The multi-year surface current fields, for COF radar (March 2012 to January 2021) and NEWC radar (November 2017

to January 2024), were generated by reprocessing data from two HF radar sites operating along the southeastern Australian continental shelf. The gap-filling method was tested and validated against data with synthetic gaps and the 2dVar method was shown to perform well. The surface current representation of the HF radar was tested based on validation from 3 types of drifters and current meter moorings over a 4-8 year period. In addition, the robust performance of the advanced 2dVar approach demonstrates its effectiveness as a valuable tool for processing long-term coastal radar measurements, allowing for a

more comprehensive investigation of the long-term trends and low-frequency modulations in the influential EAC system over extended future periods.

    The data provide invaluable insights into the complex circulation patterns and dynamics of the East Australian Shelf region. Our results showed that the reconstructed dataset from coastal radar observations is able to capture the coupled variability between the deep ocean EAC and the shelf circulation across scales ranging from tidal, seasonal to interannual scales as

well as mesoscale to submesoscale features. Despite some temporal gaps, this long, continuous dataset can shed light on the dynamics of the EAC and its effects on the regional circulation, such as shelf-slope interactions, upwelling patterns, frontal dynamics, and eddy-shedding events linked to the meandering EAC pathway. Overall, by bridging the observational gaps in HF radar data through advanced methods, this work has delivered a unique dataset for advancing our understanding of the complex EAC regime and its far-reaching impacts on the shelf environment, while also providing practical applications for

various stakeholders in the region.

## 7   Data availability

The radial velocities for each radar site are available at the Australian Ocean Data Network (AODN)https://thredds.aodn.org.au/thredds/catalog/IMOS/ACORN/catalog.html. The hourly total surface current data files are structured following the IMOS data format The reconstructed radar data are available at https://doi.org/10.5281/zenodo.13984639 (Tran, 2024b). The COF

mooring data are available at the AODN https://thredds.aodn.org.au/thredds/catalog/IMOS/ANMN/NSW/catalog.html. SVP



drifter data are available here https://data.pmel.noaa.gov/generic/erddap/tabledap/gdp_hourly_velocities.html. The reanalysis wind BARRA2 data are available at https://dx.doi.org/10.25914/1x6g-2v48.

## 8 Code availability

The variational interpolation (2dVar) approach for reprocessing HF radar data and the scripts for plotting the data are available
at https://doi.org/10.5281/zenodo.13985075 (Tran, 2024a).

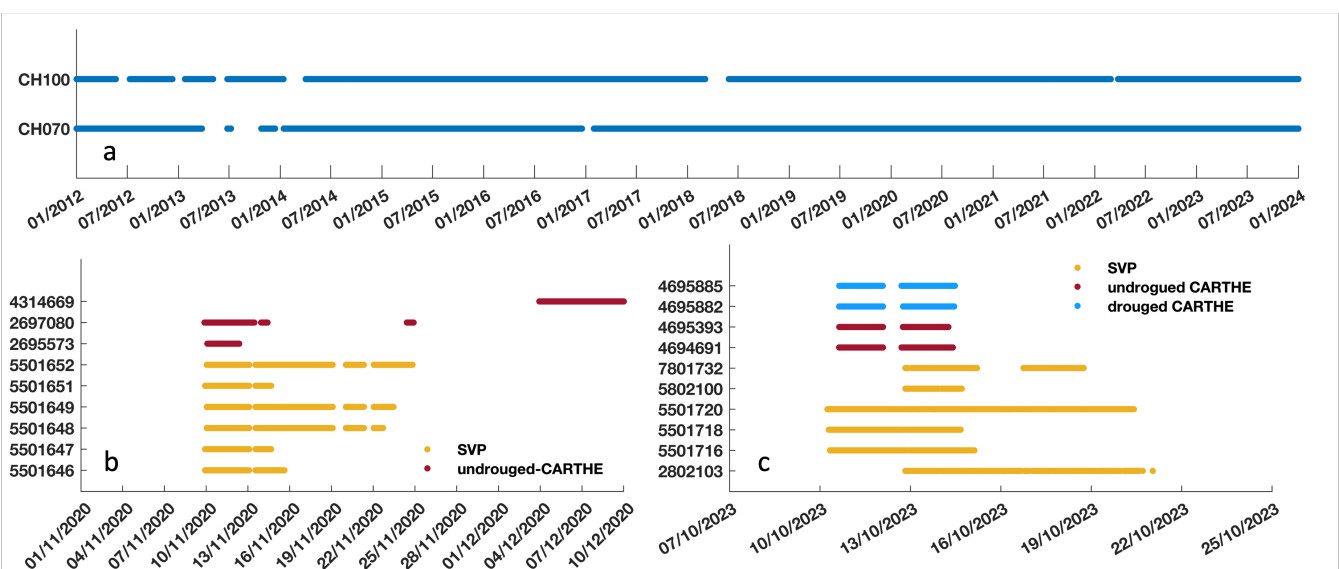

**Figure A1.** Mooring (a) and drifter data (b, c) are available for radar comparison from 01 February 2012 to 31 December 2023. The drifters, including the surface (drogued and undrogued CARTHE) and near-surface types (SVP), were deployed in two periods: one in early November 2020 (b) and the other in early October 2023 (c). Only drifter data points that remained within the HF radar domain were used. The y-axis displays the names of the observational instruments. Gaps denote missing data.

*Author contributions.*

    MCT: conceptualization, methodology, data analysis and investigation, visualizations, and writing - original draft. MR: Funding acquisition, Resources, conceptualization, methodology, supervision, writing – review, and editing. AS: conceptualization, methodology, writing – review, and editing.

*Competing interests.*


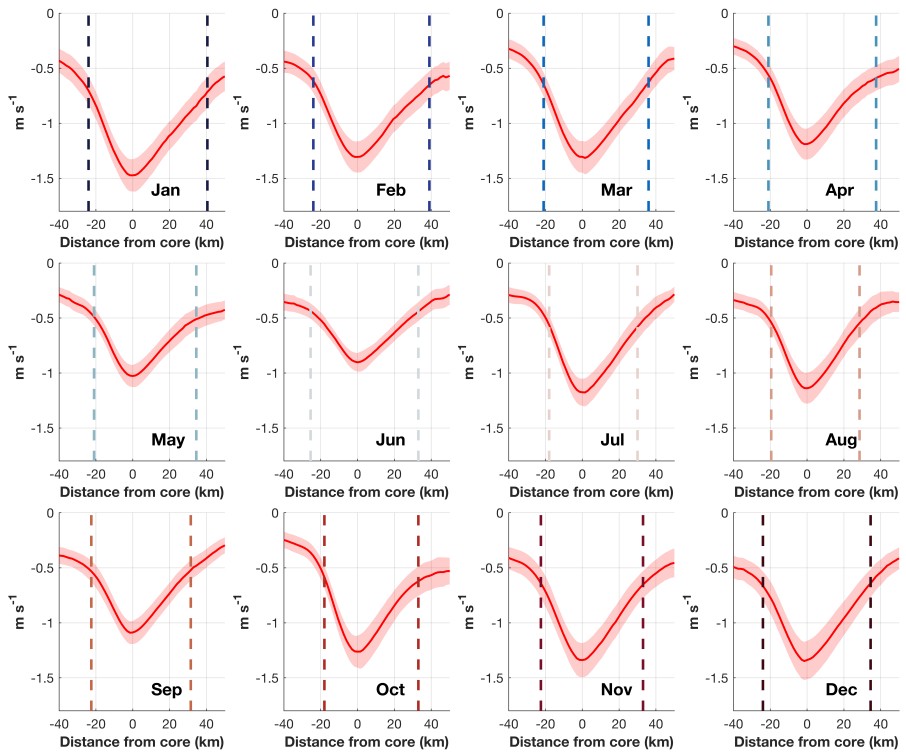

**Figure A2.** Annual cycle of the EAC cross-structure identified from the jet-following method (Archer et al., 2017b) based on 8 years of Coff Habour (COF) data. A positive value indicates the southward movement of the jet. Cross-structure of the jet is averaged 30.2° - 30.6°S. Red lines represent the mean poleward magnitude of the EAC across the jet. Shading areas represent one standard deviation from the mean speed. Dash lines mark the boundary of the EAC, defined as points with a 50% reduction from the core velocity.

The contact author has declared that none of the authors has any competing interests.

*Acknowledgements.* We acknowledge the IMOS HF Radar team for their work in the deployment, maintenance and ongoing developments, and calibration of the HF radar systems. SVP Drifters were provided by NOAA global drifter program https://www.aoml.noaa.gov/global-drifter-program/ and deployed from the RV Bombora and the RV Investigator through a grant of sea time on RV Investigator from the Marine National Facility (https://ror.org/01mae9353. HF radar and mooring data were sourced from Australia's Integrated Marine Observing System

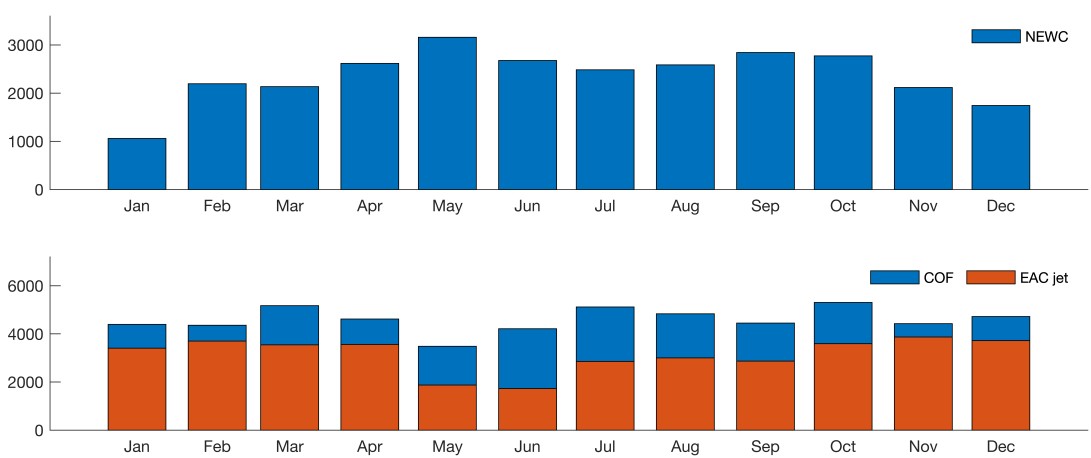

**Figure A3.** The number of hourly domain-averaged HF radar data available as the function of the month for each radar site: NEWC (top) and COF (bottom) radar. The EAC jet bars (orange) represent the times that the jet was detected by the jet-following method from the COF data.

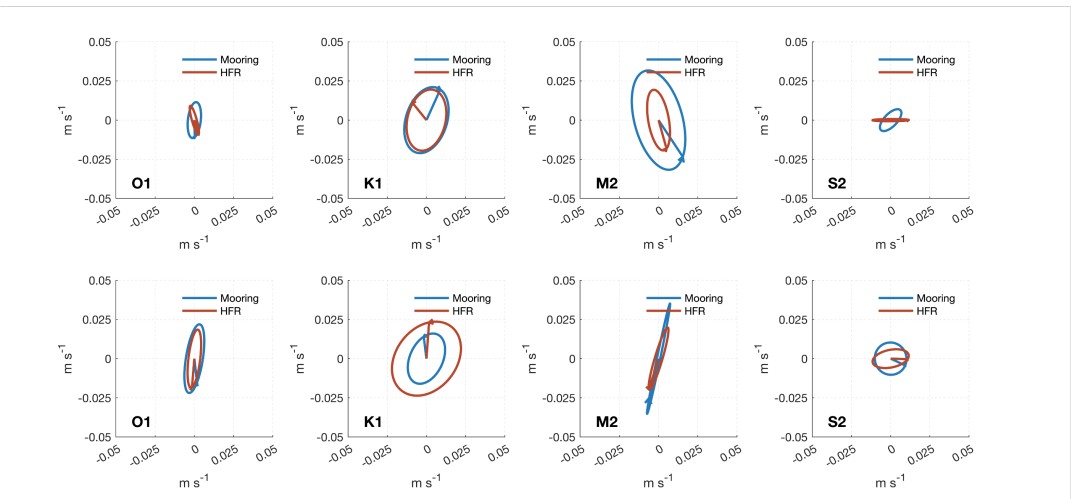

**Figure A4.** Tidal current ellipses extracted from the more than 1-year mooring data and HF radar-derived velocity at CH070 (15 November 2018 to 15 Aug 2020) and CH100 (from 15 February 2019 to 15 October 2019). The tidal phase in UTC is shown by corresponding lines.

(IMOS) – IMOS is enabled by the National Collaborative Research Infrastructure Strategy (NCRIS). This research includes computations using the computational cluster Katana supported by Research Technology Services at the University of New South Wales (UNSW Sydney) (https://doi.org/10.26190/669x-a286).



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
