# Peer review of "Surface current variability in the East Australian Current from long-term HF radar observations"

_Earth System Science Data, 2024_

## Referee Comment (RC3)

**Surface Current Variability in the EAC**

Line 2          I question the phrase decadal fluctuations, they have only recorded 8 years of data

Line 25         "intricate flow patterns"  can you describe more

Line 57         Tasman Front moves towards

Line 63         capitalize Integrated Marine Observation System

Line 79         unweighted least-squares (Lipa reference)

Lipa, B., and D. Barrick. "Least-squares methods for the extraction of surface currents from CODAR crossed-loop data: Application at ARSLOE." *IEEE Journal of Oceanic Engineering* 8, no. 4 (1983): 226-253.

Line 86         what does the (2) refer to?

Line 126        remove "land based"

Line 140        CODAR Ocean Sensors

Line 149        sentence ends with preposition from, please fix

Line 153        direction finding

Table 1         the formatting of the table could be improved.  Can you add lines for each row.  I think this would make it easier to read.

Figure 1        I think you should separate the coverage maps for the COF network and RHED network since the length of time they were operating were so different.  The colormap for Figure 1a should be changed to cmocean speed.

Line 163        what does FV00 mean, can you explain more

Figure 2        why did the coverage decrease at RRK and NNB starting in 2021

Line 189        remove sentence "The drifters will be described more …"

Line 201        Each SVP drifter

Line 243        is the equation needed?

Line 255          independent validation dataset, which is?

Line 270          comparison was

Line 303          what were the validation points

Line 308          imbalance of radar observations, please explain more

Line 352          can you explain "resulting in reversal" more

Table 3           theta phase difference, can you explain more

Table 4           can you make the gap filling 2dvar data its own section, so you'll have LS, 2dVar and 2dVar gap filled as the 3 main columns

Line 446          radar-derived currents

Figure 8          the ellipses over the vectors are hard to read, can you separate

Line 484          I thought the COF data was 8 years?

Figure 9          Can you group the panels by season summer (Dec, Jan, Feb) etc.

Line 498          the widening jet, can you explain that more, I don't see it in Fig 11

Fig 11            can you color the months by season summer (black), fall (red), winter (blue), spring (green)

Line 508          mooring and radar data

Line 535          within the range of ____ cm/s , missing number

Line 609          I disagree with the statement that the radar datasets have certain inherent limitations, if one of the radar stations goes offline, then you lose totals.  That is a limitation in the design of the observing system, not the radar technology.  Redundant coverage would eliminate this limitation.

---

## Author Response (AR1)

**Response to Reviewers (**Manuscript ID: essd-2024-480)

**Title: Surface current variability in the East Australian Current from long-term HF radar observations**

We greatly appreciate the constructive comments from the Reviewers. Below we provide our detailed point-by-point responses and any description of actions taken regarding the comments by the Reviewers.

**Reviewer 1**

In this manuscript, the authors presented analysis results of the sea surface current derived from HF radar data collected in the East Australian area. The investigation is comprehensive and the data may be useful for researchers who are interested in this area. The topic fits the journal, the manuscript is well written, I suggest the authors consider the following problems in revision:

We are pleased the reviewer has found our manuscript and data to be useful for researchers interested in the region.

Technical comments:
1. Beside studies on long-term variation, research on rapid varying current is very important (see, e.g., DOI: 10.1109/JOE.2016.2591718).
   **Answer:** Thank you for your suggestion. The rapid fluctuation of the ocean currents is indeed important. Therefore, we added text and citations in the revised manuscript in line 47.

2. You may also consider comparison of the currents results obtained by Seasonde and WERA systems.
   **Answer:** This is a good point, and has been investigated in other regions, however the two operating radar systems in the east Australia current are situated far away from each other (more than 300 km, beyond the observational range of both radars). Which prohibits this analysis in our region.

3. Add explanation about how W^u is chosen.
   **Answer:** In this work, we performed a similar practice as in Yaremchuk et al. (2017), section 4.1, for identifying the weight parameters of the 2dVar approach. The $W^u$ was roughly estimated based on the formula: $W^u = 0.05\sigma^2 l^4$, in which $\sigma^2$ is the diagonal values of the noise covariance matrix from the radial data and $l$ is the spatial resolution of the radial data. The equation represents the cut-off scale, which is approximately twice the radial resolution. After fixing the $W^u$ value, we adjust the value of $W^c$ and $W^d$ using the drifter data for NEWC and mooring data for COF radar until the optimal values were found. New text was added to the section 3.2 for clarity.

Other comments:
1. Line 245, delete "the" before "K".

**Answer:** Thank you for your suggestion. The phrase was removed in the revised manuscript.

**Reviewer 2**

This article utilizes 2D-Var and HF radar data to measure surface currents both upstream and downstream of the East Australian Current (EAC). It then analyzes the variability of the EAC across multiple spatial and temporal scales, highlighting its critical role in influencing continental shelf dynamics, regional circulation, coastal weather, and global climate patterns.

However, the description of the second term in the 2D-Var cost function J is unclear. Specifically, what is meant by "facilitating the extraction of the large-scale circulation pattern while limiting the generation of spurious small-scale variations in the reconstructed velocity field"? Why is this approach effective in achieving these outcomes?

**Answer:** The phrase was modified for clarity as "...to facilitate the smoothness of the circulation pattern while limiting the generation of spurious small-scale variations in the reconstructed velocity field" as well as the section 3.2 for describing the 2dVar approach.

The second term in the 2dVar cost function is introduced to constrain the kinematic of the flow field which was introduced by Kaplan et al. (2007). In the 2dVar algorithm, the Laplacian operator acts as a high-pass filter which can result in a more 'violent' circulation regime or noisy reconstructed field. As demonstrated by Yaremchuk and Sentchev (2009), enforcing the smoothness of the divergence and vorticity patterns is beneficial to facilitate the smoothness of the circulation pattern.

Additionally, why was 2D-Var chosen over 3D-Var or 4D-Var?

**Answer:** The 2dVar approach is used here because it was developed as an inexpensive algorithm for real-time interpolation of surface currents by a typical HFR system. Besides its ability to gap-fill data, the algorithm is simpler compared to more sophisticated methods like the open-boundary modal analysis (OMA) (Kaplan et al., 2007). The interpolation field can be simplified by adjusting three weight parameters. Additionally, the method has shown good performance in several studies, such as those in Bodega Bay (Yaremchuk and Sentchev, 2009), the Iroise Sea (Sentchev et al., 2013), and the Gulf of Tonkin (Tran et al., 2020). Therefore, we believe this method is suitable for the purpose of this study to create a long-term radar dataset in eastern Australia.

Meanwhile, new methods are being researched to enhance data accuracy, representing a promising area for future investigation.

Could the reconstruction be improved by including surface wind stress and other atmospheric variables? These could provide valuable constraints and enhance the robustness of the results.

**Answer:** The reviewer makes a good point, however, we do not have spatially resolved data at the same resolution as the radar. This could be an area for further investigation in the future.

Finally, consider exploring broader climate connections. For example, how does EAC variability relate to larger climate phenomena, such as the El Niño-Southern Oscillation (ENSO) or the Indian Ocean Dipole (IOD)? Additionally, assessing the impact of EAC variability on regional ecosystems, fisheries, and biodiversity—particularly in the context of climate change—could provide important insights and expand the study's relevance.

**Answer:** The reviewer makes a good point, and this dataset will be useful for further investigation – however it is outside the scope of this data description paper to answer these science questions here.

**Reviewer 3**

We greatly appreciate the constructive comments from the Reviewer. Below we provide our detailed point-by-point responses and any description of actions taken regarding the comments by the Reviewer.

Line 2 I question the phrase decadal fluctuations, they have only recorded 8 years of data

**Answer:** Thank you for pointing this out. In this context, we referred to the variability of the EAC as observed in previous studies (not the results of this study). The EAC has been an important subject for study for a long period, e.g., seasonal, interannual (Sloyan et al., 2016), decadal (Hill et al., 2011), and long-term (Oliver et al., 2015), etc. For clarity, we have changed the phrase decadal fluctuations to long-term fluctuations which we believe that is more suitable.

Line 25 "intricate flow patterns" can you describe more

**Answer:** We have added the text **"**(e.g. frontal eddies and filaments)" here.

Line 57 Tasman Front moves towards

**Answer:** We have corrected the text as follows:
"here part of the jet separates from the coast, shedding eddies that flow eastward forming the EAC eastern extension of and those that continue southward form the EAC southern extension"

Line 63 capitalize Integrated Marine Observation System

**Answer:** We have corrected the text as required.

Line 79 unweighted least-squares (Lipa reference) Lipa, B., and D. Barrick. "Least-squares methods for the extraction of surface currents from CODAR crossed-loop data: Application at ARSLOE." IEEE Journal of Oceanic Engineering 8, no. 4 (1983): 226-253.

**Answer:** Thank you for this suggestion. We have added the citations to the revised manuscript and made the reference to the text.

Line 86 what does the (2) refer to?

**Answer:** Thank you for pointing this out.  This was a typo and was removed from the revised manuscript.

Line 126 remove "land based"
**Answer:** The phrase was removed in the revised manuscript.

Line 140 CODAR Ocean Sensors
**Answer:** The phrase was corrected in the revised manuscript.

Line 149 sentence ends with preposition from, please fix
**Answer:** The sentence was rewritten as follows: "As the transmitted radio wave is reflected back to the radar from all directions, the WERA-manufactured radar uses the beam-forming method to determine the position of the signal".

Line 153 direction finding
**Answer:** The phrase was corrected in the revised manuscript.

Table 1 the formatting of the table could be improved. Can you add lines for each row. I think this would make it easier to read.
**Answer:** We have modified the Table 1 as required. Lines were added in each row for clarity. However, we note this may be changed by the journal at the typesetting stage.

Figure 1 I think you should separate the coverage maps for the COF network and RHED network since the length of time they were operating were so different.
**Answer:** Thank you for your comment. Indeed, the coverage maps in Fig. 1b were plotted separately for each radar site during their operating periods: COF radar was from 01 March 2012 to 01 January 2021 and NEWC radar from 01 November 2017 to 01 January 2024.
New text has been added to clarify this point in the Fig. 1 caption for better clarification as follows: "Maps of the mean spatial coverage (as a ratio from 0 to 1) were plotted separately for the two sites; COF (01 March 2012 - 01 January 2021) and NEWC (01 January 2018 - 01 January 2024)"

The colormap for Figure 1a should be changed to cmocean speed.
**Answer:** We acknowledge that the cmocean delta was indeed an odd choice for plotting the ocean current maps. However, the high contrast between low and high values in the cmocean delta allows for better visualization of the EAC and the coastal region than the speed colormap. Here, we empirically chose the value of 0.6 m s$^{-1}$, which is demonstrated by the white band in this colormap, as the boundary of the EAC. In this way, the strong and weak current regions are well separated by the blue and green color bands, respectively. The benefit of using the cmocean delta colormap is the EAC can be well recognized in the radar-derived current maps, as shown in Fig. 1 as well as in Fig. 3, Fig. 8, and Fig. 9.

Line 163 what does FV00 mean,
**Answer:** FV00 here refers to the radar real-time products with a basic quality controlled which are publicly available on the AODN website (https://thredds.aodn.org.au/thredds/catalog/IMOS/ACORN/catalog.html). The more enhanced data (FV01) data is provided at the delay of few months (Cosoli and Grcic,

2019). In our study, the FV01 data for the NEWC were only available for two years 2018 and 2019, therefore, we had to apply the QC to our NEWC radar data as the guidance from Cosoli and Grcic (2019).

The text was modified for clarity as follows: "In the real-time product, an IMOS standard quality control procedure was applied to remove the data outliers from the original radial data (FV00). A more comprehensive quality control (QC) procedure is then applied to the FV00 data to create a more accurate product, which is published to the AODN server after a delay of a few months and flagged as FV01 as per (Cosoli and Grcic, 2019)"

**can you explain more Figure 2 why did the coverage decrease at RRK and NNB starting in 2021**
**Answer:** Since 2021, the Red Rocks (RRK) and North Nambucca (NNB) radars have experienced several hardware issues related to antennas, cables, hardware, and site computers. These issues affected radar operations, reducing the coverage of both radars. Consequently, both radar sites were partially or non-operational for the past three years. This has been added to the text.

**Line 189 remove sentence "The drifters will be described more …"**
**Answer:** The phrase was removed in the revised manuscript.

**Line 201 Each SVP drifter**
**Answer:** The phrase was corrected in the revised manuscript.

**Line 243 is the equation needed?**
**Answer:** We also questioned whether including a complex equation in the manuscript was useful. However, since this is a data description paper, the method for creating the data should be clear. Therefore, we decided to leave the equation in the manuscript to give the readers a general view of the 2dVar approach used in this study. Additionally, we provide a sample script along with this paper for those who are interested.

**Line 255 independent validation dataset, which is?**
**Answer:** The validation dataset was the cross-validation points that set aside from the radial data for each of the radar site. In total, there were respectively about 1,160 and 1,189 snapshots for NEWC and COF radar. From there, roughly around 2% of total points (for example 550,599 points in total of 27,529,950 from 1,160 snapshots in NEWC) were chosen to compare with reconstructed values.

To validate the method, the total velocities reconstructed from the remained data were interpolated on those cross-validation points. The radial velocities computed from the reconstructed total velocities were then compared with the original dataset that set aside in the first step. The results of comparison are shown in Table 2 with different scenarios of data gaps.

To avoid confusion, we have also revised the sentences as follows: "These analyzed current vectors were then interpolated onto the locations corresponding to the cross-validation points, facilitating the evaluation of the accuracy of the analysis"

**Line 270 comparison was**

**Answer:** The phrase was corrected in the revised manuscript.

Line 303 what were the validation points

**Answer:** This is also in line with the above question in Line 255. The RMSE in Fig. 4 was computed from the comparison at every cross-validation point that set aside at the beginning of the validation. The result in Fig. 4 was averaged over 1,160 snapshots for the NEWC radar.

Line 308 imbalance of radar observations, please explain more

**Answer:** We believe this is a good point for discussion. The imbalance of radar observations here refers to the number of radial data from both sites when they are used to combine a total velocity in the single snapshot. For the radial balance distribution test in this study, we applied the guidance from IMOS procedure (Cosoli and Grcic, 2019) in which the ratio of the radial observations between site 1 (NOBS1) and site 2 (NOBS2) are equal (NOBS1 = NOBS2) or not significantly different (1 < NOBS1/NOBS2 < 10).

The uncertainty of the total velocity can be attributed to multiple factors, such as errors in radial data, the number of radial data points used, and the Geometrical Dilution of Precision (GDOP), etc. Indeed, the analysis domain for the NEWC region was carefully chosen to lie within an area of good GDOP values, minimizing errors as much as possible (Fig. 3c, d). Additionally, Fig. 4 shows high RMSEs (> 15 cm s$^{-1}$) along the radial beam of the northern radar site, while lower RMSEs (5–8 cm s$^{-1}$) were found for the southern site. Normally, we would expect the northern radar site to have better accuracy since the mean circulation indicated that the EAC pathway is parallel to the northern radial beam (Fig. 1a). Initially, we thought the imbalance in radial data was responsible for the high error in total velocity reconstruction. However, the local circulation in this region is also influenced by the energetic, large-scale circulation, causing rapid changes in the local circulation regime (e.g., Fig. 4 in Malan et al. (2023)). This led us to believe that the large errors at the northern site in the offshore region were not only due to the imbalance of radar observations but caused by multiple factors, including radar uncertainties, the location of the northern sites, and the complex dynamics of the region.

For clarity we have improved the text as follows: "Note that higher velocity errors are found off the region of the SEAL radar site toward the offshore region. This significant offshore discrepancy coincides with the highly energetic region of the EAC pathway (Fig. 3c, d). The local circulation in this region is also influenced by the energetic, large-scale circulation, causing rapid changes in the local circulation regime (e.g., Fig. 4 in Malan et al. (2023)). The large errors in the offshore region were likely due to multiple factors, including radar uncertainties, the location of the northern sites, and the complex dynamics of the region."

Line 352 can you explain "resulting in reversal" more

**Answer:** The texts have been revised for clarity as follows: "The lack of a drogue in the CARTHE drifter increases its sensitivity to Stokes drift (Novelli et al., 2017), causing the

offshore CARTHE to closely follow the wind direction (Fig. 5d). This behavior contrasts with that of the SVP drifter and radar-derived current vectors. (Fig. 5e)"

Table 3 theta phase difference, can you explain more

**Answer:** Based on the analysis with $\theta$ values, we noticed that the deviation of the radar-derived and drifter vectors for both methods (LS and 2dVar) was quite satisfied, about 5 degrees. A large deviation of $\theta$ was found for the near-shore group deployed in 2020 (-3.9 and 4.1 for LS and 2dVar, respectively) which we believed was due to the drifters travelling close to the baseline between two radars. Along with the good correlation and RMSE values, it convinced us to use the method for reconstructing the radar dataset. Other than that, the $\theta$ contains the information about the dynamics of the surface (CARTHE, ~ 0 – 0.4 m) and subsurface (radar, ~2.4 m and SVP, ~15 m). In general, we found a more consistent of $\theta$ for radar-derived and SVP vectors (Table 4) during two campaigns, which generally agrees with the Ekman theory about rotation of current from surface to depth. The $\theta$ values between CARTHE drifters and radar-derived vectors, however, were varied between two campaigns (e.g., -0.9 in 2020 and 8.1 in 2023 for undrouge CARTHE and 2dVar current vectors). This variation can be related to the wind force since most of our drifters used in this study were the undrouged drifters, which were more sensitive to the surface wind. The behavior of the drifters and the dynamics of the near-surface layer was our point of discussion in the section 5.1.

Table 4 can you make the gap filling 2dvar data its own section, so you'll have LS, 2dVar and 2dVar gap filled as the 3 main columns

**Answer:** Thank you for your suggestion. Table 4 had been revised.

Line 446 radar-derived currents

**Answer:** The phrase was corrected in revised manuscript.

Line 484 I thought the COF data was 8 years?

**Answer:** In this section, we referred to the work of Archer et al. (2017a), who used COF radar data to analyse EAC dynamics. However, only four years of COF radar data (2012 to 2016) were used for the Archer analysis. To avoid confusion, we have modified the text: "Archer et al. (2017a), using the jet-following method and COF radar data from 2012 to 2016, revealed that the EAC magnitude and its associated variance follow a seasonal pattern, peaking during summer".

Figure 8 the ellipses over the vectors are hard to read, can you separate

**Answer:** See next comment

Figure 9 Can you group the panels by season summer (Dec, Jan, Feb) etc.

**Answer:** We have modified the Fig. 8 and Fig. 9 in the revised manuscript. Following the suggestion, we have reduced the intensity of the ellipses and increased the ellipse scale for clarity. The monthly mean currents maps are now sorted in each column which represents the austral summer to spring.

[Figure]

Figure 8. Maps showing the monthly mean radar-derived current vectors at Coffs Harbour (upstream) using hourly data from the COF radar from March 2012 to January 2021. The monthly mean currents maps are organized by seasons in each column. The velocity unit is m s⁻¹. The current velocity variances are illustrated by plotting ellipses at 6-grid point intervals for visualization. The bathymetry contours are plotted at 100, 200, 1000, 2000, 3000 and 4000 m levels.

[Figure]

Figure 9. Maps showing the monthly mean radar-derived current vectors off Newcastle using hourly data from the NEWC radar from November 2017 to February 2024. The monthly mean currents maps are organized by seasons in each column. The velocity unit is m s⁻¹. The current velocity variances are illustrated by plotting ellipses at 2-grid

point intervals for visualization. The bathymetry contours are plotted at 100, 200, 1000, 2000, 3000 and 4000 m levels.

Fig 11 can you color the months by season summer (black), fall (red), winter (blue), spring (green)
Line 498 the widening jet, can you explain that more, I don't see it in Fig 11
**Answer:** Fig. 11 has been revised. Now the mean EAC core velocities for each month were plotted separately for clarity.

[Figure]

Figure 11. Annual cycle of the EAC cross-structure identified from the jet-following method (Archer et al., 2017b) based on 8 years of Coff Habour (COF) data. A positive value indicates the southward movement of the jet. Cross-structure of the jet is averaged 30.2∘ - 30.6∘S. Red lines represent the mean poleward magnitude of the EAC across the jet. Shading areas represent one standard deviation from the mean speed. Dash lines mark the boundary of the EAC, defined as points with a 50% reduction from the core velocity.

Line 508 mooring and radar data
**Answer:** The phrase was corrected in the revised manuscript.

Line 535 within the range of ____ cm/s , missing number

**Answer:** We apologize for this mistake, the values were added and the sentence was corrected as follows: "Kirincich et al. (2019) compared the 25 MHz HF radar in the Martha's Vineyard with the CODE drifter found the RMSE within the range of 5 to 10 cm s⁻¹ in the center while the error up to 20 cm s⁻¹ were found at the outer edge of the radial coverage and correlation of around 0.73".

Line 609 I disagree with the statement that the radar datasets have certain inherent limitations, if one of the radar stations goes offline, then you lose totals. That is a limitation in the design of the observing system, not the radar technology. Redundant coverage would eliminate this limitation

**Answer:** Thank you for your suggestion. We agree with this. The text was removed from the revised manuscript.

**Editor notes:**

1. It seems, that the text "(IMOS) – IMOS is enabled by the National Collaborative Research Infrastructure Strategy (NCRIS). This research includes computations using the computational cluster Katana supported by Research Technology Services at the University of New South Wales (UNSW Sydney) https://doi.org/10.26190/669x-a286" on the page 34 should belong to some section of the manuscript. With the next revision, please move the text to the appropriate section (if necessary).

**Answer:** This has been moved to the acknowledgements

2. Since each DOI link (alternative: review link or other means of data access), no matter where, must be accompanied by an "in-text" citation (e.g., Wagner et al., 2020), I kindly ask you re-check whether can DOI links https://doi.org/10.26190/669x-a286, https://dx.doi.org/10.25914/1x6g-2v48 be accompanied by citations. If yes, please add "in-text" citations to these DOIs and add their full citations to the section "References".

**Answer:** We have corrected these links to the BARRA2 data and UNSW Katana computing resource.